# Contextually Affinitive Neighborhood Refinery for Deep Clustering

Chunlin Yu[1]    Ye Shi[1,2]    Jingya Wang[1,2*]

[1] ShanghaiTech University
[2] Shanghai Engineering Research Center of Intelligent Vision and Imaging
{yuchl,shiye,wangjingya}@shanghaitech.edu.cn

## Abstract

Previous endeavors in self-supervised learning have enlightened the research of deep clustering from an instance discrimination perspective. Built upon this foundation, recent studies further highlight the importance of grouping semantically similar instances. One effective method to achieve this is by promoting the semantic structure preserved by neighborhood consistency. However, the samples in the local neighborhood may be limited due to their close proximity to each other, which may not provide substantial and diverse supervision signals. Inspired by the versatile re-ranking methods in the context of image retrieval, we propose to employ an efficient online re-ranking process to mine more informative neighbors in a Contextually Affinitive (ConAff) Neighborhood, and then encourage the cross-view neighborhood consistency. To further mitigate the intrinsic neighborhood noises near cluster boundaries, we propose a progressively relaxed boundary filtering strategy to circumvent the issues brought by noisy neighbors. Our method can be easily integrated into the generic self-supervised frameworks and outperforms the state-of-the-art methods on several popular benchmarks. Code is available at: https://github.com/cly234/DeepClustering-ConNR.

## 1 Introduction

Fueled by the expressive power of neural networks, deep clustering has emerged as a prominent solution in the low-label regime, where traditional clustering can be significantly enhanced by utilizing latent data representations. Recent advancements in self-supervised learning have further expanded the possibilities for deep clustering [38, 17, 41], providing a solid foundation for grouping similar instances. Among the various self-supervised recipes, contrastive learning methods ensure instance discrimination backed up with sufficient negative samples, while non-contrastive learning methods circumvent the class collision issue [37, 17] by simply pulling two augmented views closer. However, capturing the intrinsic semantic structure with clear decision boundaries remains a challenge in deep clustering, which has led researchers to focus on more group-aware and discriminative approaches.

Group-aware methods commonly enforce the prediction consistency between samples and their nearest neighborhoods, measured by cosine similarity or Euclidean distance for deep clustering[42, 21, 55, 56]. Although these methods are simple and effective, a significant portion of the retrieved neighborhoods may contain semantically redundant information. This is primarily due to the fact that the local neighboring samples are already close to the anchor sample in the feature space, and as such, may not provide sufficient supervision signals for deep clustering. To address this limitation, previous works have added memory banks with historical features to enlarge the search space [21, 49], or labeled the proximity between samples over the entire dataset [55, 56]. Building on the foundation of

---

*Corresponding author.

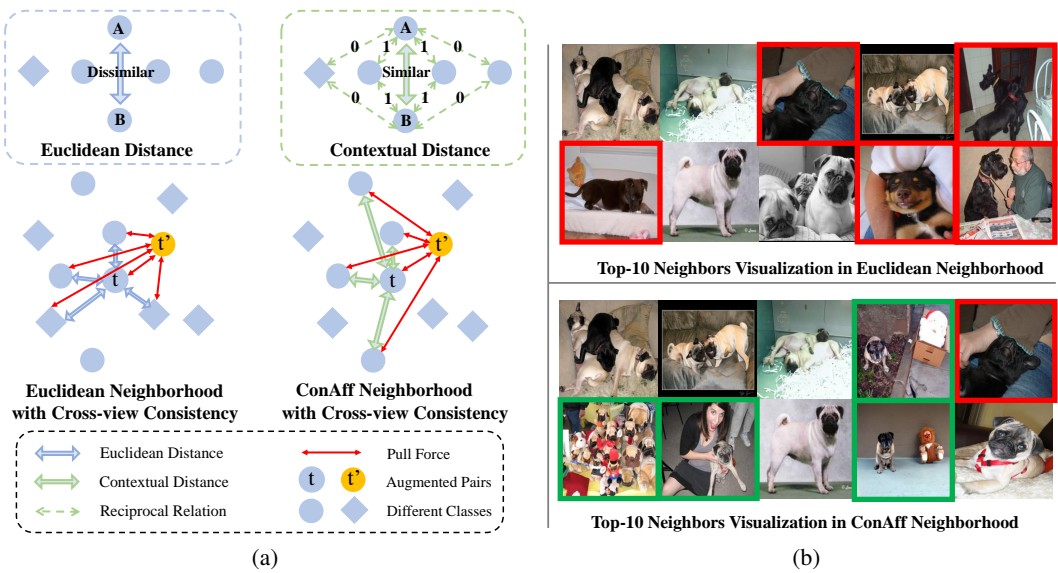

Figure 1: (a) Comparison between the conventional Euclidean Neighborhood and the ConAff Neighborhood, both with cross-view consistency. The Euclidean distance retrieves the Euclidean neighborhood, while the contextual distance is used for the ConAff neighborhood. In reciprocal relations, "1" means two samples are in each other's top-$k$ neighbors, and "0" indicates neither is in the other's top-$k$. The goal is to use reciprocal relations as a contextual distance metric for the ConAff neighborhood. For instance, distant pairs A and B might be contextually similar due to their similar reciprocal relations. (b) Images in the first column and their 10 nearest neighbors in the other columns, were retrieved using Euclidean Neighborhood and the proposed ConAff Neighborhood. Wrong neighbors are marked in red, and hard positives are marked in green. By default, unmarked neighbors are regarded as true neighbors.

previous group-aware methods, a key research question arises: *Can we excavate more informative and reliable neighbors from intra-batch samples?*

Inspired by the versatile re-ranking techniques that have shown remarkable performance in image retrieval tasks [57, 45, 32, 44, 45], we propose a novel approach to deep clustering that involves retrieving neighborhoods in a contextually affinitive metric space. Typically, the neighborhood information is retained in the ranking list of pairwise similarities between a query sample and other samples. However, such a list may neglect the contextual statistics that exist between the samples. When two images are contextually affinitive, they may share similar relationships to a set of reference images. In this study, the reference images are defined as the current batch, and the relationship between two images is established based on whether they are among each other's top-$k$ neighbors. Consequently, distant image pairs in the original feature space may be contextually affinitive, and therefore considered as more informative positive pairs. Additionally, the contextually affinitive neighborhood in one view may align with the query sample in the other view, thus enabling cross-view neighborhood consistency. Figure 1 (a) presents a comparison between the conventional Euclidean Neighborhood and the proposed ConAff Neighborhood. Additionally, Figure 1 (b) illustrates the results obtained when retrieving the nearest neighbors using both types of neighborhoods.

In the context of learning with neighborhoods, intrinsic noises can exist that are less observable compared to external noises caused by label contamination. These noises originate from the inferiority of the intrinsic feature space and taking these noisy samples as candidates for clustering can lead to the accumulation errors and degrade overall performance. To address this challenge, we propose a progressive boundary filtering strategy based on the ConAff neighborhood that further improves the robustness of clustering. Our strategy filters out boundary samples in a self-paced learning paradigm and gradually involves more complex samples for clustering.

Our contributions are summarized as follows:

- We propose a **Co**ntextually affinitive **N**eighborhood **R**efinery framework (CoNR) for deep clustering, which attempts to excavate more semantically relevant neighbors in a contextually affinitive space, termed as ConAff neighborhood.

- We devise a progressive boundary filtering strategy to combat the intrinsic noises existing in the ConAff neighborhood, further improving the robustness of clustering.

- Extensive experiments demonstrate that CoNR improves upon the SSL baseline by a significant margin, proving to be competitive with state-of-the-art methods on five widely-used benchmarks. CoNR's simplicity and effectiveness allow it to be easily integrated into other SSL frameworks, resulting in consistent performance gains.

## 2 Methodology

Our goal is to group a dataset containing $n$ unlabeled samples into $k$ semantic clusters, where $k$ refers to the cardinality of class label sets. Since it is non-trivial to directly obtain clean cluster assignments from scratch, we adopt an unsupervised pre-text task as the initialization for image clustering, which provides a decent neighborhood structure. However, directly utilizing the neighboring information could lead to sub-optimal results as i) the nearest-neighbor searching schema based on cosine similarity may not favor hard positives that are important for clustering, and ii) the top-ranked neighbors of samples located at a cluster boundary contain inevitable noise when the learned clusters have not been not mutually separated. We, therefore, propose our main algorithm for clustering in a contextually affinitive neighborhood, with progressive relaxation of boundary sample filtering, in an attempt to alleviate the above issues.

### 2.1 Preliminary

In general, given a dataset $\mathcal{D} = \{\boldsymbol{x}_1, \boldsymbol{x}_2, \cdots, \boldsymbol{x}_n\}$, an image $\boldsymbol{x}_i$ uniformly sampled from $\mathcal{D}$, and two augmentations $t \sim \mathcal{T}, t' \sim \mathcal{T}'$, we aim to first encourage the instance-aware concordance based on BYOL [11] for early pre-training, and then encourage the group-aware concordance for later finetuning, in which the cornerstone of our method lies. Following [11], the model is equipped with an online network $q(\mathcal{F}_o(\cdot))$ and a target network $\mathcal{F}_t(\cdot)$, where each of $\mathcal{F}_o(\cdot), \mathcal{F}_t(\cdot)$ consists of an encoder cascaded by a projector, and $q(\cdot)$ constitutes an additional non-linear predictor.

#### 2.1.1 Instance-aware Concordance

The commonly-used instance discrimination approach for non-contrastive self-supervised learning enforces the similarity between a referenced feature and its corresponding feature in another augmented view, which can be formulated as:

$$\mathcal{L}_{sim}^I = - \mathop{\mathbb{E}}_{\boldsymbol{x}_i \in \mathcal{B}} \left[ \langle q(\mathcal{F}_o(t(\boldsymbol{x}_i)), \mathcal{F}_t(t'(\boldsymbol{x}_i))) \rangle \right], \tag{1}$$

where $\langle \cdot \rangle$ denotes the cosine similarity. This can be interpreted as the promotion of instance-aware concordance under different augmentations and hidden transformations, which could help bootstrap an over-clustered representation via the stop-gradient and momentum-update techniques. Notably, this instance-aware concordance produces a general-purpose representation that can be well transferred to various downstream tasks, as well as a solid basis for neighborhood discovery and higher-order contextual exploration.

#### 2.1.2 Group-aware Concordance

In order to develop the discrimination framework from instance level to group-aware concordance, we consider regarding the local neighborhood as a semantic group that involves more diverse positive samples, sharing the same spirit with previous works [10, 21, 42, 29]. Typically, group-aware concordance can be formulated as:

$$\mathcal{L}_{sim}^G = - \mathop{\mathbb{E}}_{\boldsymbol{x}_i \in \mathcal{B}} \left[ \mathop{\mathbb{E}}_{\boldsymbol{x}_j \in \mathcal{N}(\boldsymbol{x}_i)} \left[ \langle q(\mathcal{F}_o(t(\boldsymbol{x}_i)), \mathcal{F}_t(t'(\boldsymbol{x}_j))) \rangle \right] \right], \tag{2}$$

where $\mathcal{N}(\boldsymbol{x}_i)$ denotes the set of neighborhood samples of $\boldsymbol{x}_i$ preserved in a Euclidean metric space, and $\boldsymbol{x}_i \in \mathcal{B}$ reveals that all samples in the batch are uniformly considered to encourage the cross-view

neighborhood consistency. While the de-facto methods [10, 21, 42, 29] exploit neighborhoods to enrich the latent representation, they somehow ignore the quality of neighborhoods themselves. On the one hand, good neighbors should be clean, which means that we should select proper candidates from $\mathcal{B}$ to retrieve clean neighbors. On the other hand, good neighbors should be rich, which motivates us to replace $\mathcal{N}(\boldsymbol{x}_i)$ with more informative neighborhoods in higher-order space. Therefore, we seek to further improve the group-aware concordance paradigm from two perspectives, which will be introduced in Sec. 2.2 and Sec. 2.3.

## 2.2 Contextually Affinitive Neighborhood Discovery

As discussed in Sec. 2.1, previous works retrieve the nearest neighbors based on pairwise cosine similarity, where the distance metric depends solely on two involved data points on the hypersphere. On the contrary, our solution is to redefine how two data points are similar by considering contextual relations among the whole batch data. Concretely, we first transform the data points into a contextually affinitive space, where we assume that the more similar two transformed data points are, the more high-order contextual information they share, and thus we can utilize this information to excavate more meaningful neighbor pairs, which we term as the contextually affinitive neighborhood discovery.

Formally, given a batch of normalized features $\mathcal{B}_f = [\boldsymbol{f}_0, \cdots, \boldsymbol{f}_{|\mathcal{B}_f|-1}]^T$, our goal is to first transform $\mathcal{B}_f$ into a contextually refined feature space $\mathcal{H}_f = [\boldsymbol{h}_0^r, \cdots, \boldsymbol{h}_{|\mathcal{B}_f|-1}^r]^T$, and then retrieve the nearest neighbors set using the distance metric defined by $\mathcal{H}_f \mathcal{H}_f^T$. Inspired by [53, 57, 34] which build the contextual relations over the entire dataset, we efficiently retrieve the contextually affinitive neighborhood via topology graphs in an online manner following [53].

**Building Contextual $k$-NN Graph**. The underlying principle of building the contextual $k$-NN graph is to inject contextual information into nodes $\mathcal{V}$ and edges $\mathcal{E}$ so that the message propagation can move beyond the local region [53]. First, an online reciprocal adjacency matrix $\boldsymbol{A} \in \mathbb{R}^{|\mathcal{B}_f| \times |\mathcal{B}_f|}$ is computed as follows:

$$\boldsymbol{A}_{ij} = \begin{cases} 1, & \text{if } \boldsymbol{f}_i \in \mathcal{N}(\boldsymbol{f}_j, k_1) \wedge \boldsymbol{f}_j \in \mathcal{N}(\boldsymbol{f}_i, k_1) \\ 0, & \text{if } \boldsymbol{f}_i \notin \mathcal{N}(\boldsymbol{f}_j, k_1) \wedge \boldsymbol{f}_j \notin \mathcal{N}(\boldsymbol{f}_i, k_1) \\ 0.5, & \text{otherwise} \end{cases}, (3)$$

where $\boldsymbol{f}_j \in \mathcal{N}(\boldsymbol{f}_i, k_1)$ denotes $\boldsymbol{f}_j$ is among the $k_1$ nearest neighbors of $\boldsymbol{f}_i$ in $\mathcal{B}_f$, and each row of $\boldsymbol{A}$ can be interpreted as the $k$-reciprocal encoding [57]. Then, the graph $G = (\mathcal{V}, \mathcal{E})$ can be defined as:

$$\mathcal{V} = \{v_i | v_i = [\boldsymbol{A}_{i,0}, \cdots, \boldsymbol{A}_{i,|\mathcal{B}_f|-1}], i \in \{0, \cdots, |\mathcal{B}_f|-1\}\}, \quad (4)$$

$$\mathcal{E} = \{e_{ij} | e_{ij} = \langle \boldsymbol{f}_i, \boldsymbol{f}_j \rangle, \boldsymbol{f}_j \in \mathcal{N}(\boldsymbol{f}_i, k_2), i \in \{0, \cdots, |\mathcal{B}_f|-1\}\}, \quad (5)$$

where $\boldsymbol{f}_j \in \mathcal{N}(\boldsymbol{f}_i, k_2)$ denotes that $\boldsymbol{f}_j$ lies within the $k_2$ nearest neighbors of $\boldsymbol{f}_i$ in $\mathcal{B}_f$.

**Message Propagation**. After building the contextual relational graph, the message propagation can further obtain high-order relations among $k$-reciprocal encoding acquired at the previous stage, similar to the query expansion technique [35].

$$\boldsymbol{h}_i^{(l+1)} = \boldsymbol{h}_i^{(l)} + \text{aggregate}(\{e_{ij}^\alpha \cdot \boldsymbol{h}_j^{(l)} | e_{ij} \in \mathcal{E}\}), \quad (6)$$

where $\alpha$ is a fixed value to realize the $\alpha$-weighted query expansion ($\alpha$-QE), and $e_{ij} \in \mathcal{E}$ ensures that the most confident $k_2$ nearest neighbors are selected for aggregation. Normally, $k_2$ should be far less than $k_1$. For clarity, if we assume the number of layers for message propagation is $L$, the initial input for the message propagation stage, $\boldsymbol{h}_i^{(0)}$, is derived from $v_i$ from the previous stage. After propagation across all $L$ layers, the output $\boldsymbol{h}_i^{(L)}$ becomes our desired contextually refined feature, represented as $\boldsymbol{h}_i^r$.

**Affinitive Neighborhood Retrieval**. So far, the intra-batch data points have been mapped to a contextual metric space via a two-stage transformation. The underlying ranking statistics within these transformed data points should be more accurate and informative. Therefore, we reveal the hidden properties by explicitly retrieving the affinitive neighborhood using the refined node features, which we term as the contextually affinitive neighborhood discovery. Concretely, given the original features $[\boldsymbol{f}_0, \cdots, \boldsymbol{f}_{|\mathcal{B}_f|-1}]^T$ and the refined features $[\boldsymbol{h}_0^r, \cdots, \boldsymbol{h}_{|\mathcal{B}_f|-1}^r]^T$, we define the affinitive neighborhood $\mathcal{N}^a(i, k)$ as:

$$\mathcal{N}^a(i, k) = \{j | \boldsymbol{h}_j^r \in \mathcal{N}(\boldsymbol{h}_i^r, k)\}. \quad (7)$$

Then the group-aware discrimination can be reformulated as:

$$\mathcal{L}_{sim}^{GA} = - \mathop{\mathbb{E}}_{\boldsymbol{x}_i \in \mathcal{B}} \big[ \mathop{\mathbb{E}}_{j \in \mathcal{N}^a(i,k)} [\langle q(\mathcal{F}_o(t(\boldsymbol{x}_i)), \mathcal{F}_t(t'(\boldsymbol{x}_j))\rangle]\big]. \tag{8}$$

We remark that the contextually affinitive neighborhood discovery ensures more informative neighbors are involved for training at every iteration, and the superior contextual neighbor information lying between the non-differentiable transformed data points is synchronously propagated to the original features to improve the representation quality.

## 2.3 Progressive Boundary Filtering with Relaxation

In this section, we are focused on the inevitable noise existing in the neighborhood. Although the contextually affinitive neighborhood could offer more contextually relevant samples to boost training, there still exist some unreliable samples among the top-ranked neighbors. Such a phenomenon becomes more evident when the sampled data points lie on the decision boundary. As a consequence, the inclusion of contaminated neighbor information will cause noise accumulation, to the detriment of later training iterations. Therefore, we propose an online boundary sample filtering with progressive relaxation to alleviate the issue.

### 2.3.1 Online Boundary Sample Detection

In order to better assess the current clustering quality and detect the samples located at the boundary, we apply k-means after each training epoch, which is a common practice in former works [17, 38]. Instead of directly utilizing the consistency of pseudo labels among neighbors, we place more emphasis on the holistic cluster structure, which is to distinguish farther samples located at the cluster boundary with controlled degrees. Inspired by the silhouette score which measures the ratio between the mean intra-cluster distance and the mean nearest-cluster distance in an offline manner, we devise an approximate online version to serve a similar purpose.

Formally, given a sample $\boldsymbol{x}_i$ with pseudo label $\boldsymbol{\pi}(\boldsymbol{x}_i)$, and $K$ cluster centroids $\boldsymbol{C} = [\boldsymbol{c}_1, \cdots, \boldsymbol{c}_K]^T$ obtained from last epoch, we first approximate the mean intra-cluster distance $d_I$ and nearest mean inter-cluster distance $d_N$ by:

$$d_i^I = \|\boldsymbol{f}_i - \boldsymbol{c}_{\boldsymbol{\pi}(\boldsymbol{x}_i)}\|_2, \quad d_i^N = \|\boldsymbol{f}_i - \boldsymbol{c}_{\boldsymbol{\pi}'(\boldsymbol{x}_i)}\|_2, \tag{9}$$

where

$$\boldsymbol{\pi}'(\boldsymbol{x}_i) = \mathop{\arg\min}_{k \in \{1, \cdots, K\} \backslash \{\boldsymbol{\pi}(\boldsymbol{x}_i)\}} \|\boldsymbol{f}_i - \boldsymbol{c}_k\|_2, \tag{10}$$

And then the boundary ratio can be computed as:

$$r_i = 1 - \frac{d_i^N - d_i^I}{\max(d_i^I, d_i^N)}. \tag{11}$$

Normally, a large boundary ratio indicates the given sample is more likely to be located at the boundary, while a small ratio implies the sample is close to its cluster center, and it is safer to exploit non-boundary samples for training since they exhibit cleaner neighborhoods.

### 2.3.2 Progressive Relaxation

While boundary samples may possess inferior neighborhoods, it is barely true to ignore these samples during the entire clustering process. To make a balance, we progressively relax the restrictions of samples to promote neighborhood consistency. The motivations are two folds. First, we set a strict criteria to select clean samples at the initial stage to avoid the confirmation bias brought by noisy samples. Second, after the neighborhoods have been consistently refined using cleaner samples, the boundary samples can further improve the overall clustering quality without affecting the holistic structure, which can be validated in our ablation experiments.

Formally, given a batch $\mathcal{B}$ sampled from the dataset, we adopt a linearly increased fraction ratio $fr^{(t)}$ to dynamically control a filtering threshold $\sigma^{(t)}$ which is used to filter out the hard boundary samples at epoch $t$.

$$fr^{(t)} = fr^{(t_0)} + \frac{1 - fr^{(t_0)}}{t - t_0} * (T - t_0), \tag{12}$$

---

**Algorithm 1** The proposed algorithm CoNR

---

**Require:** Dataset $\mathcal{D}$, online network $\theta$, and target network $\hat{\theta}$

1: Pre-train $\theta$, $\hat{\theta}$ on $\mathcal{D}$                                                             ▷ Eq. (1)
2: **while** Clustering **do**
3:     Sample batch $\mathcal{B}$ from $\mathcal{D}$
4:     Transform batch features $\mathcal{B}_f$ to $\mathcal{H}_f$
5:     Retrieve contextually affinitive neighborhood $\mathcal{N}^a$ for $\mathcal{B}_f$ using $\mathcal{H}_f$          ▷ Eq. (7)
6:     Progressively filter out boundary samples for clustering               ▷ Eq. (13)
7:     Update $\theta$ with $\mathcal{L}_{sim}^{GAF}$ and SGD optimizer                       ▷ Eq. (14)
8:     Update $\hat{\theta}$ with momentum moving average
9: **end while**

---

where $t$ is the current epoch, $t_0$ is the epoch to start clustering, and $T$ is the maximum number of epochs for group-aware concordance clustering. Then $\sigma^{(t)}$ can be obtained by retrieving the largest ratio among the top $|\mathcal{B}| * fr^{(t)}$ smallest boundary ratios in $\{r_0, \cdots, r_{|\mathcal{B}|-1}\}$. According to the $\sigma^{(t)}$, we filter out these samples whose boundary ratios are too large to be involved in clustering at the current epoch $t$:

$$\mathcal{B}^{(t)} = \{\boldsymbol{x}_i | r_i \leq \sigma^{(t)}, i \in \{0, \cdots, |\mathcal{B}|-1\}\}. \tag{13}$$

Notably, our progressive boundary filtering strategy is essentially different from the self-labeling step that selects the most confident samples for fine-tuning in prior work [42, 29]. First, their selection strategy is completely based on prediction confidence or pseudo-label consistency, neglecting the intrinsic cluster structure to evaluate the difficulty of samples. Second, their self-labeling step is based on cross-entropy loss, which heavily relies on stronger augmentations to avoid overfitting. By contrast, our filtering strategy is proposed to search a robust neighborhood and avoid accumulation errors during the group-aware concordance, which does not require additional modules or augmentations.

## 2.4 The Overall Objective and Optimization

Based on Sec. 2.2 and Sec. 2.3, our final objective for group-aware concordance can be further formulated as:

$$\mathcal{L}_{sim}^{GAF} = - \mathbb{E}_{\boldsymbol{x}_i \in \{\mathcal{B} \backslash \mathcal{B}^{(t)}\}} \big[ \mathbb{E}_{j \in \mathcal{N}^a(i,k)} [\langle q(\mathcal{F}_o(t(\boldsymbol{x}_i)), \mathcal{F}_t(t'(\boldsymbol{x}_j))\rangle]\big], \tag{14}$$

where $\boldsymbol{x}_i \in \{\mathcal{B} \backslash \mathcal{B}^{(t)}\}$ controls whether the samples should be selected as candidates to retrieve neighborhood, and $j \in \mathcal{N}^{(a)}(i,k)$ determines whether the samples are contextually affinitive to the candidates. We do not exclude $\mathcal{B}^{(t)}$ from $\mathcal{N}^{(a)}(i,k)$ as our aim is to filter out unreliable candidates lying around the boundaries, which means that for each selected candidate, we take its ConAff neighborhood as a whole to preserve its original structure. The overall training pipeline is shown in Algorithm 1.

As discussed in Sec. 2.1, our framework comprises two training stages: instance-aware concordance and group-aware concordance, and the final objective for optimization can be expressed as:

$$\mathcal{L}_{total} = \mathbb{I}(t < t_0)\mathcal{L}_{sim}^I + \mathbb{I}(t \geq t_0)\mathcal{L}_{sim}^{GAF} \tag{15}$$

# 3 Related Work

## 3.1 Deep Clustering

Deep clustering, an integration of deep learning and traditional clustering, has emerged as a de facto paradigm to learn feature representations and cluster unlabelled data simultaneously. Originated from seminal works [48], deep clustering has seen a paradigm shift from traditional methodologies [6, 36, 28, 43, 52] to more sophisticated models [4, 12, 14, 15, 30, 39]. The advent of contrastive learning in deep clustering [23, 24, 38] has brought new perspectives but also introduced challenges, mainly in defining contrastive losses and balancing instance pairs. Concurrently, self-supervised learning and multi-modal clustering techniques are pushing the boundaries further, with hybrid models combining generative and discriminative aspects, leading to richer representations and improved performance in deep clustering.

## 3.2 Self-Supervised Learning

Historically, self-supervised learning (SSL) approaches for representation learning have primarily utilized generative models [9] or relied on uniquely crafted pretext tasks such as solving jigsaw puzzles [31] and colorization [51] to the general-purpose representations. However, the emergence of contrastive learning methods [13, 5, 47] has proven highly effective for both representation learning and succeeding tasks. Despite their effectiveness, these methods necessitate a vast array of negative examples to ensure instance-aware discrimination in the embedding space. Apart from instance discrimination, group-aware discrimination focuses more on the semantic structure. While Deep Clustering [2], ODC [50] and COKE [33] disentangle the clustering stage and the learning stage, SeLa [1] and SwAV [3] attempt to solve the grouping problem via optimal transport. Another line of methods [21, 10, 27] focuses on neighborhood consistency, consistently bootstrapping the representation by discovering abundant semantically similar instances.

## 3.3 Re-ranking in Image Retrieval

Image Retrieval aims to retrieve the gallery images that are the most similar to the query image from a large corpus of images. Re-ranking is a training-free technique that is used to reorder and improve the initial ranking result using higher-order similarity metrics. Generally, $k$-NN-based re-ranking represents the fashion in image re-ranking. $k$-NN-based re-ranking views the $k$-reciprocal nearest neighbors of an image as highly relevant candidates [54, 34, 20]. Jegou *et al.* [18] iteratively correct distance estimates based on local vector distributions, Liu *et al.*[25] utilize graph convolutional networks (GCN) to encode neighbor information directly into image descriptors. Zhong *et al.* [57] encodes the reciprocal information of a query image into a vector and computes similarities using Jaccard distance. Zhang *et al.* [53] accelerate [57] by using $k$-reciprocal encodings as the node features of a GNN and enhancing the features via graph propagation. Specifically, our ConAff neighborhood is inspired by [53, 57] which considers contextual similarity with query expansion.

# 4 Experiments

## 4.1 Datasets and Settings

Following [17, 23], we report the deep clustering results on five widely-wised benchmarks, including CIFAR-10 [22], CIFAR-20 [22], STL-10 [7], ImageNet-10 [4], ImageNet-Dogs [4]. CIFAR-10 and CIFAR-20 both contain 60,000 images, and CIFAR-20 uses 20 superclasses from CIFAR-100 following prior practice [17, 23]. STL contains 100,000 unlabeled images and 13,000 labeled images. ImageNet-10 selects 10 classes from ImageNet-1k, containing 13,000 images while ImageNet-Dogs selects 15 different breeds of dogs from ImageNet-1k, containing 19,500 images. For image size, we use 32x32 for CIFAR-10 and CIFAR-20, 96x96 for STL-10 and ImageNet-10, 224x224 for ImageNet-Dogs, following the prior work [17]. For the dataset split, both train and test data are used for CIFAR-10 and CIFAR-20, both labeled and unlabeled data are used for STL-10, and only training data of ImageNet-10 and ImageNet-Dogs are used, which is strictly the same setting with [17, 38, 23, 24]. Also, all the experiments are conducted with a known cluster number.

## 4.2 Implementations

For data augmentations, we strictly follow [17], which uses ResizedCrop, ColorJitter, Grayscale, and HorizontalFlip, for a fair comparison with previous works [17, 38, 41]. For loss computation, we use a symmetric loss by swapping the two augmentations and computing the asymmetric losses twice. For architecture, we use ResNet-18 for small-scale datasets CIFAR-10 and CIFAR-20 and ResNet-34 for the other datasets, following [23, 17, 40]. For CIFAR-10 and CIFAR-20, we replace the first convolution filter of size 7x7 and stride 2 with a filter of size 3x3 and stride 1, and remove the first max-pooling layer, following [17]. All datasets are trained with 1000 epochs, where the first 800 epochs are trained with standard BYOL loss $\mathcal{L}_{sim}^{I}$, and the remaining 200 epochs are trained with our proposed $\mathcal{L}_{sim}^{GAF}$. We adopt the stochastic gradient descent (SGD) optimizer and the cosine decay learning rate schedule with 50 epochs of warmup. The base learning rate is 0.05 with a batch size of 256. For instance-aware concordance, we directly follow [11, 17] to establish a fair baseline. For group-aware concordance, we set $k, k_1, k_2$ to 20,30,10 for ImageNet-Dogs and 10,10,2 for other datasets, since ImageNet-Dogs is a fine-grained dataset compared with previous ones.

Table 1: Clustering result comparison (in percentage %) with the state-of-the-art methods on five benchmarks.

| | CIFAR-10 | | | CIFAR-20 | | | STL-10 | | | ImageNet-10 | | | ImageNet-Dogs | | |
|---|---|---|---|---|---|---|---|---|---|---|---|---|---|---|---|
| | NMI | ACC | ARI | NMI | ACC | ARI | NMI | ACC | ARI | NMI | ACC | ARI | NMI | ACC | ARI |
| IIC [19] | 51.3 | 61.7 | 41.1 | - | 25.7 | - | 43.1 | 49.9 | 29.5 | - | - | - | - | - | - |
| DCCM [46] | 49.6 | 62.3 | 40.8 | 28.5 | 32.7 | 17.3 | 37.6 | 48.2 | 26.2 | 60.8 | 71.0 | 55.5 | 32.1 | 38.3 | 18.2 |
| PICA [16] | 56.1 | 64.5 | 46.7 | 29.6 | 32.2 | 15.9 | - | - | - | 78.2 | 85.0 | 73.3 | 33.6 | 32.4 | 17.9 |
| SCAN [42] | 79.7 | 88.3 | 77.2 | 48.6 | 50.7 | 33.3 | 69.8 | 80.9 | 64.6 | - | - | - | - | - | - |
| NMM [8] | 74.8 | 84.3 | 70.9 | 48.4 | 47.7 | 31.6 | 69.4 | 80.8 | 65.0 | - | - | - | - | - | - |
| CC [23] | 70.5 | 79.0 | 63.7 | 43.1 | 42.9 | 26.6 | 76.4 | 85.0 | 72.6 | 85.9 | 89.3 | 82.2 | 44.5 | 42.9 | 27.4 |
| MiCE [41] | 73.7 | 83.5 | 69.8 | 43.6 | 44.0 | 28.0 | 63.5 | 75.2 | 57.5 | - | - | - | 42.3 | 43.9 | 28.6 |
| GCC [56] | 76.4 | 85.6 | 72.8 | 47.2 | 47.2 | 30.5 | 68.4 | 78.8 | 63.1 | 84.2 | 90.1 | 82.2 | 49.0 | 52.6 | 36.2 |
| TCL [24] | 81.9 | 88.7 | 78.0 | 52.9 | 53.1 | 35.7 | 79.9 | 86.8 | 75.7 | 87.5 | 89.5 | 83.7 | 62.3 | 64.4 | 51.6 |
| IDFD [40] | 71.1 | 81.5 | 66.3 | 42.6 | 42.5 | 26.4 | 64.3 | 75.6 | 57.5 | 89.8 | 95.4 | 90.1 | 54.6 | 59.1 | 41.3 |
| TCC [38] | 79.0 | 90.6 | 73.3 | 47.9 | 49.1 | 31.2 | 73.2 | 81.4 | 68.9 | 84.8 | 89.7 | 82.5 | 55.4 | 59.5 | 41.7 |
| ProPos [17] | 85.1 | 91.6 | 83.5 | 58.2 | 57.8 | 42.3 | 75.8 | 86.7 | 73.7 | 89.6 | 95.6 | 90.6 | 73.7 | 77.5 | **67.5** |
| DivClust [26] | 72.4 | 81.9 | 68.1 | 44.0 | 43.7 | 28.3 | - | - | - | 89.1 | 93.6 | 87.8 | 51.6 | 52.9 | 37.6 |
| BYOL [11] | 79.4 | 87.8 | 76.6 | 55.5 | 53.9 | 37.6 | 71.3 | 82.5 | 65.7 | 86.6 | 93.9 | 87.2 | 63.5 | 69.4 | 54.8 |
| CoNR (Ours) | **86.7** | **93.2** | **86.1** | **60.4** | **60.4** | **44.3** | **85.2** | **92.6** | **84.6** | **91.1** | **96.4** | **92.2** | **74.4** | **79.4** | 66.7 |

## 4.3 Comparison with the State-of-the-Art

**Comparison Settings**. In this section, we compare our methods CoNR with the current state-of-the-art methods, as shown in Table 1. Specifically, IIC, DCCM, and PICA are clustering methods without contrastive learning. SCAN and NMM are multi-stage methods that require pre-training and fine-tuning. DivClust is a recently proposed method based on concensus clustering by controlling the degree of diversity, and we report their best performance among different diversity levels in Table 1 for comparison. The other methods mostly learn representations based on contrastive learning or non-contrastive learning in an end-to-end fashion. For these methods that learn a generic representation rather than directly output cluster assignments, we use $k$-means for evaluation. In particular, for ProPos, BYOL, and CoNR (ours), the produced features output by the target encoder are used for $k$-means for a fair comparison.

**Comparison Results**. Regarding the quantitative results, we achieve consistent performance gains across the board. Specifically, for relatively small datasets such as ImageNet-10 and STL-10 with 13K images, our method reaches satisfying results, which not only surpass our baseline BYOL by at least +10.1% ACC on STL-10 in only 200 epochs, but also further achieve a new state-of-the-art performance with 96.4% ACC on ImgeNet-10. For moderate-scale datasets such as CIFAR-10 and CIFAR-20 with 60K images, our method yields stable performance improvement with an average gain of +1.6% and +2.6% ACC compared with ProPos [17] on CIFAR-10 and CIFAR-20, respectively. More importantly, for the challenging fine-grained dataset ImageNet-Dogs, we achieve +6.2% and +1.9% performance boosts in terms of ACC on our baseline and the most advanced competitors. We ascribe this significant improvement to the superiority of our ConAff neighborhood, especially on the fine-grained task. Additionally, we provide results on Tiny ImageNet and performance comparisons in Appendix A.3.

## 4.4 Ablation Study

In this section, we perform a comprehensive ablation experiment in Table 2 to validate the effectiveness of our proposed method. Specifically, InC represents the baseline, CnAffN, OBD, PR are techniques proposed by our method, and LN is a standard choice used for performance comparison.

**Validation of Contextually Affinitive Neighborhood**. We make the following observations: First, our proposed contextually-affinitive neighborhood (ConAffN) consistently outperforms the local neighborhood approach, irrespective of whether a progressive boundary filtering strategy is utilized or not. However, ConAffN achieves its peak performance when both online boundary filtering and progressive relaxation are utilized. Instead, the local neighborhood meets with performance saturation when combined with progressively relaxed boundary filtering. Also, as illustrated by the purity curve in Figure 2, ConAffN identifies more reliable neighbors, particularly when the value of K is larger.

**Validation of Progressive Boundary Filtering with Relaxation**. As previously discussed, both LN and ConAffN combined with OBD experience consistent performance gains, with +1.9% and +1.4%,

Table 2: Ablation Results (in percentage %) on CIFAR-10. (InC: Instance-aware Concordance, LN: Local Neighborhood, CnAffN: Contextually Affinitive Neighborhood, OBD: Online Boundary Detection, PR: Progressive Relaxation)

| InC | LN | CnAffN | OBD | PR | CIFAR-10 | | |
|---|---|---|---|---|---|---|---|
| | | | | | NMI | ACC | ARI |
| ✓ | | | | | 79.4 | 87.8 | 76.6 |
| ✓ | ✓ | | | | 81.9 | 89.6 | 78.7 |
| ✓ | | ✓ | | | 84.6 | 91.1 | 82.4 |
| ✓ | ✓ | | ✓ | | 83.9 | 91.1 | 82.3 |
| ✓ | ✓ | | ✓ | ✓ | 84.8 | 91.6 | 84.2 |
| ✓ | | ✓ | ✓ | | 85.7 | 92.5 | 85.1 |
| ✓ | | ✓ | ✓ | ✓ | 86.7 | 93.2 | 86.1 |

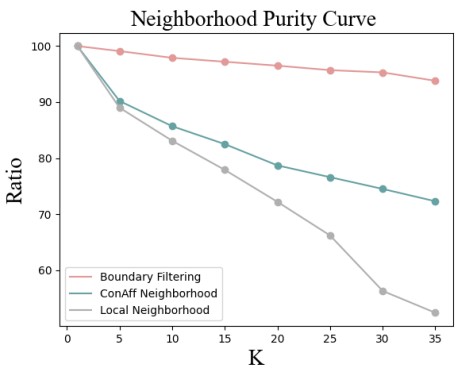

Figure 2: The curve of the purity (in percentage %) with the changes of K.

respectively. We speculate that the boundary filtering strategy uniformly improves the robustness of neighborhood clustering. Moreover, the online boundary detection exhibits a much cleaner neighborhood (Figure 2), which supports our motivation. Despite the efficacy of OBD, we find progressive relaxation (PR) drives the model towards less biased clustering as it gradually involves the difficult samples for clustering, which further improves the ACC by +0.7%.

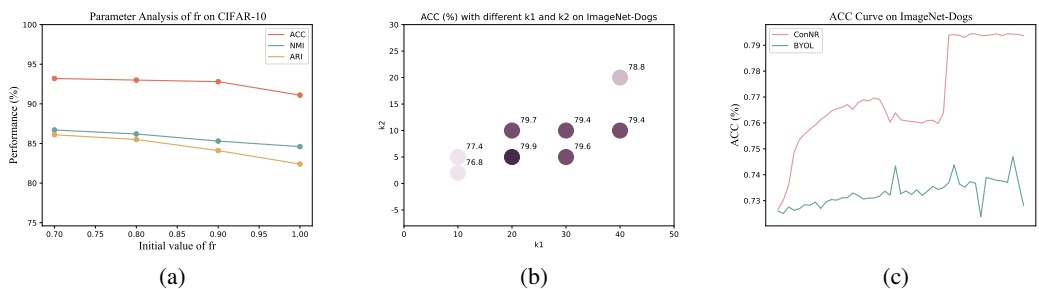

(a) (b) (c)

Figure 3: (a) Performance comparison with different initial fraction ratios on CIFAR-10. (b) Performance with a different selection of $k_1$, $k_2$ on ImageNet-Dogs. (c) Clustering performance comparison with ConNR and BYOL on ImageNet-Dogs.

**More Discussions**. We conducted an analysis of three parameters: $k_1$ and $k_2$, which represent the number of neighbors used to construct the graph, and $fr$, the fraction ratio set at the initial training stage. Figure 3(a) presents the results with varying $fr$ values. The outcomes exhibit relative stability within the range of [0.70, 0.80, 0.90] for $fr$, however, a noticeable performance decline is observed when $fr = 1$. This indicates a lack of boundary filtering applications. We provide the results with different choices of $k_1$ and $k_2$ in Figure 3(b). For non-aggressive strategy where $k_1$ and $k_2$ are smaller than 10, there are consistent gains compared with the baseline, while for more aggressive choices where $k_1$, $k_2$ are larger, the performance can be improved even further, validating the design of contextually affinitive neighborhood. In Figure 3(c), we present the ACC curves generated during the clustering process. Here, we can see that ConNR not only converges considerably faster than BYOL, but also provides notable improvements when implemented as an add-on module. A detailed exploration of how ConNR benefits other self-supervised benchmarks is provided in Appendix A.2.

## 5 Conclusion and Limitations

In this paper, we proposed a novel method to improve deep clustering in self-supervised learning by promoting the semantic structure preserved by neighborhood consistency. Our approach, the Contextually Affinitive (ConAff) Neighborhood, employs an efficient online re-ranking process to mine more informative neighbors and encourages cross-view neighborhood consistency. We also

introduced a progressively relaxed boundary filtering strategy to mitigate the intrinsic neighborhood noises near cluster boundaries. Our method outperformed state-of-the-art methods on several popular benchmarks and could be easily integrated into generic self-supervised frameworks. One potential limitation of our approach is that ConNR generally assumes the labels induce an equipartition of the whole data and thus does not contain specific mechanisms for handling unbalanced or long-tailed datasets. We believe that our approach can be further extended for this purpose. While our major goal in this paper is dedicated to learning a clustered and well-separated representation for deep clustering, the proposed ConAff neighborhood s a generic design choice that could be viewed from a broader perspective, hopefully benefiting the task of self-supervised learning that focuses more on downstream applications.

## Acknowledgement

This work was supported by NSFC (No.62303319), Shanghai Sailing Program (21YF1429400, 22YF1428800), Shanghai Local College Capacity Building Program (23010503100), Shanghai Frontiers Science Center of Human-centered Artificial Intelligence (ShangHAI), MoE Key Laboratory of Intelligent Perception and Human-Machine Collaboration (ShanghaiTech University), and Shanghai Engineering Research Center of Intelligent Vision and Imaging.

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
