# Appendices: Contextually Affinitive Neighborhood Refinery for Deep Clustering

## A More Experimental Results

### A.1 Training Efficiency

Table 3: Comparison of the average training speed (it/s) using batch size 256 on a single 3090 RTX GPU across different methods on five benchmarks (it/s: iterations per second).

| Method | CIFAR-10 | CIFAR-20 | STL-10 | ImageNet-10 | ImagNet-Dogs |
|---|---|---|---|---|---|
| BYOL | 7.74 | 7.63 | 5.11 | 5.39 | 5.33 |
| ConNR(Ours) | 7.68 | 7.58 | 5.04 | 5.36 | 5.28 |

We show the training efficiency of ConNR by comparing its training speed with a standard efficient SSL baseline BYOL. As shown in Table 3, our method ConNR has a slightly slower training speed, however, introduces no additional heavy computational burden and is comparably efficient with the state-of-the-art methods. In fact, we ascribe the efficiency of ConNR to the highly parallelized cuda implementation and the online manner of ConAff neighborhood discovery, incurring negligible time overhead in stark contrast to conventional methods taking re-ranking as an offline post-processing technique over the entire dataset.

### A.2 Migration to Other Self-supervised Frameworks

Table 4: Performance of ConNR migrated to other self-supervised learning frameworks. ConNR* is simply notated as an add-on module to distinguish itself from ConNR based on BYOL.

| Method | CIFAR-10 | | | CIFAR-20 | | |
|---|---|---|---|---|---|---|
| | NMI | ACC | ARI | NMI | ACC | ARI |
| SimSiam | 78.6 | 85.6 | 73.6 | 52.2 | 48.5 | 32.7 |
| SimSiam + ConNR* | 85.2 (+6.6) | 91.4 (+4.8) | 83.6 (+10.0) | 59.4 (+7.2) | 59.2 (+10.7) | 43.1 (+10.4) |
| MoCo v2 | 66.9 | 77.6 | 60.8 | 39.0 | 39.7 | 24.2 |
| MoCo v2 + ConNR* | 81.4 (+14.5) | 88.9 (+11.3) | 79.1 (+18.3) | 52.6 (+13.6) | 54.5 (+14.8) | 37.7 (+13.5) |

In this section, we underline that our method **ConNR can be actually viewed as a plug-in-and-play module**, which can be easily integrated into other contrastive and non-contrastive self-supervised learning frameworks with slight modifications. Then we will detail the implementation of the integration of ConNR* to SimSiam and MoCo v2. Specifically, ConNR* refers to the modules without considering the BYOL baseline.

#### A.2.1 SimSiam + ConNR*

Since SimSiam simply encourages the similarity of two augmented features without any negative samples, following the notation of [5], the loss function of SimSiam + ConNR* can be expressed as:

$$\frac{1}{2} \cdot \frac{1}{|\mathcal{N}^a(\boldsymbol{z}_2)|} \sum_{\boldsymbol{z}_2^i \in \mathcal{N}^a(\boldsymbol{z}_2)} \mathcal{D}(\boldsymbol{p}_1, \text{stop-grad}(\boldsymbol{z}_2^i)) + \frac{1}{2} \cdot \frac{1}{|\mathcal{N}^a(\boldsymbol{z}_1)|} \sum_{\boldsymbol{z}_1^i \in \mathcal{N}^a(\boldsymbol{z}_1)} \mathcal{D}(\boldsymbol{p}_2, \text{stop-grad}(\boldsymbol{z}_1^i)), \quad (16)$$

where $\mathcal{N}_{\boldsymbol{z}_1}^a$ is the ConAff neighborhood retrieved by $\boldsymbol{z}_1$. In this way, the contextual knowledge of ConAff neighborhood can be injected into the group-aware concordance loss.

### A.2.2 MoCo v2 + CoNR*

Similarly, referring to the original loss function in MoCo v2 [13], we can generalize the loss function into CoNR* version, which can be formally represented as:

$$\mathcal{L} = -\frac{1}{|\mathcal{N}^a(k_+)|} \sum_{k_+^i \in \mathcal{N}^a(k_+)} \log \frac{\exp(q \cdot k_+^i / \tau)}{\sum_{j=0}^{K} \exp(q \cdot k_j / \tau)} \tag{17}$$

The results are shown in Table 4 with significant performance gains on both SimSiam and MoCo v2. Specifically, our method improves SimSiam and MoCo v2 by at least +4.8% and +11.3% w.r.t ACC on CIFAR-10, respectively. The flexible feasibility and evident performance gains validate the effectiveness of our design.

### A.3 Results on Large-scale Datasets

Previous experiments are based on moderate-scale datasets following the state-of-the-arts [17, 38, 40], where the total number of classes is limited. Here, we testify our method on a large-scale dataset Tiny-ImageNet which consists of 200 classes with 10,0000 training images in total. The results in Table 5 indicate that our approach can successfully scale to large datasets. Specifically, we observed a notable increase of +2.7% in NMI and +3.2% in ARI, surpassing the performance of state-of-the-art techniques. These outcomes demonstrate the effectiveness and scalability of our proposed method when applied to Tiny-ImageNet. The enhancements achieved in NMI and ARI highlight the superiority of our approach in boostrapping the underlying structures via ConAff neighborhood consistency and progressive boundary sample filtering.

Table 5: Performance comparison on Tiny-ImageNet.

| Method | Tiny-ImageNet | | |
|---|---|---|---|
| | NMI | ACC | ARI |
| DCCM | 22.4 | 10.8 | 3.8 |
| PICA | 27.7 | 9.8 | 4.0 |
| CC | 34.0 | 14.0 | 7.1 |
| GCC | 34.7 | 13.8 | 7.5 |
| TCL | 43.5 | 30.6 | 15.2 |
| BYOL | 36.5 | 19.9 | 10.0 |
| ProPos | 40.5 | 25.6 | 14.3 |
| ConNR (Ours) | **46.2** | **30.8** | **18.4** |

## B Relations to Existing Deep Clustering Methods

Although there are some representative prior works [14, 15, 42] that encourage group-aware concordance, proving to be effective in deep clustering, our proposed method extends the current paradigm by further exploiting neighborhoods in a contextually affinitive (ConAff) metric space rather than the metric space defined by cosine similarity or euclidean distance. We underline this makes our method ConNR fundamentally different from previous works. Moreover, our method proposes a progressively relaxed boundary filtering strategy to efficiently filter out unreliable candidates and relax the constraints in later iterations. Importantly, both the ConAff neighborhood and the filtering strategy consider the efficiency of implementation in a totally online manner. Additionally, our extended approach proves to be more effective when compared to the vanilla method maintaining Euclidean neighborhood consistency, as validated in our ablation experiments in the main manuscript.

## C Visualizations of Boundary Sample Detection

To provide a more intuitive understanding of how the boundary selection strategy is conducted and why it is beneficial, we add a visualisation of feature representation before and after using boundary filtering. As can be observed in Figure 4(a), the overlapped regions of the 2D cluster boundaries

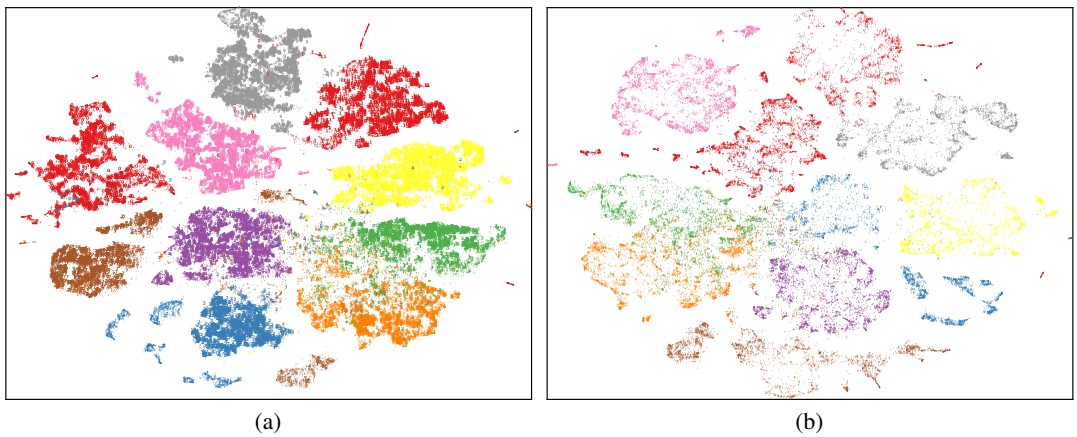

(a)                                                                    (b)

Figure 4: (a) T-SNE visualizations of all samples in CIFAR-10, where boundary samples are shown as small dots, non-boundary samples are shown as large dots. (b) T-SNE visualizations of boundary samples in CIFAR-10, where boundary samples are shown as small dots.

predominantly consist of our identified boundary samples. To provide a more focused view, Figure 4(b) exclusively displays these detected boundary samples, effectively highlighting the contours of the clusters.

## D   More Visualizations of Neighborhoods

We provide more visualizations of top-10 neighborhoods on ImageNet-10 using the checkpoints pre-trained with BYOL. As seen in Figure 5, we could draw a general conclusion that the Euclidean neighborhood struggles to capture the subtle differences between intra-class objects, while the ConAff neighborhood is more capable of grouping instances of the same class together. More specifically, we observe that airliners are the most common classes that could easily treat other classes as their neighbors, such as airships and container ships. Admittedly, these three classes resemble each other, making the Euclidean neighborhood hard to distinguish them. However, our proposed ConAff neighborhood, as shown in the fifth and sixth row of Figure 5, can better deal with the subtle differences in most cases. Interestingly, in the fourth row, the Euclidean neighborhood mistakenly treats the soccer ball held by a person and a Maltese dog held by a person as the same class, which focuses more on human-object interaction but lacks the crucial details of objects. By contrast, the ConAff neighborhood manages to disentangle the dogs from their owners, demonstrating the robustness of eliminating the background noises.

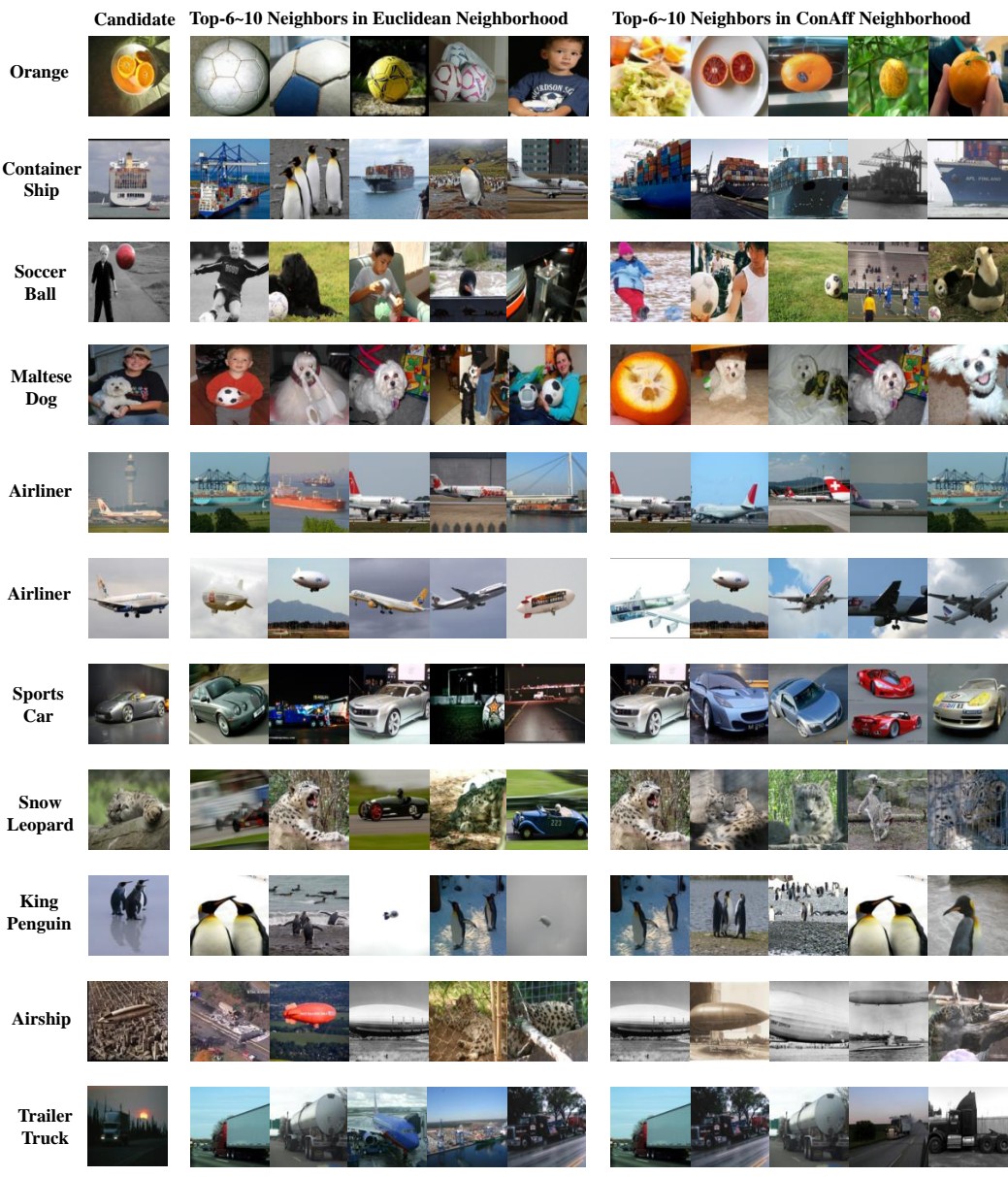

Figure 5: More visualizations of top-10 neighborhood on ImageNet-10.