# OpenReview forum: "Contextually Affinitive Neighborhood Refinery for Deep Clustering"
_NeurIPS.cc/2023/Conference — NeurIPS 2023 poster_

### Official Review · Reviewer_vt5R · 2023-06-28

**Soundness:** 4 excellent
**Presentation:** 4 excellent
**Contribution:** 3 good
**Rating:** 7
**Confidence:** 4

**Summary:**

This paper proposes a simple yet effective method to learn class-sensitive image feature representations by exploring local neighbourhood structures, which are shown discriminative and beneficial to image clustering with the help of off-the-shelves clustering algorithms like k-means. This is achieved by mining "rich" (less visually redundant to the anchors) neighbours in a contextual metric space embedding high-order relations among $k$-reciprocal encoding of samples. A boundary sample detection strategy is also introduced to avoid the distractions from false positive sample pairs involved in the discovered neighbourhoods. The effectiveness of the proposed model was validated on 5 standard image clustering benchmark datasets.


**Strengths:**

+ The paper is generally well-written with clear motivations and sufficient discussions of the rationale behind each model design
+ The proposed model is simple yet effective, which presents a new state-of-the-art for image clustering even though it doesn't explicitly learn the target cluster assignment as conventional deep clustering approaches

**Weaknesses:**

+ The paper is generally well-written, however, there are still rooms to improve:
    - I'd suggest clarifying at an early stage that the outputs of the proposed model are the feature representations of images rather than their cluster assignments to one of $k$ classes (L70) given that the paper claimed to propose a ``clustering'' model, which is a bit confusing to me until the experiment section mentions that K-means is adopted.
    - Is $h^{(0)}$ defined before it is used in Eq.(6)? I failed to find the definition and assumed it is $v_i$ in Eq.(4). Please correct me if I'm wrong.
    - Is $r$ in Eq.(7) the same as the bounary ratio discussed in Eq.(11)?
    - How many times the message propagation is conducted for refining the features to obtain $h^r$?
+ Given that the key idea of the proposed model is to benefit global data structure modelling (clustering) by exploring local data structure (local neighbourhoods), such an idea is closely related to "Unsupervised Deep Learning by Neighbourhood Discovery" (ICML'19) and "Unsupervised deep learning via affinity diffusion" (AAAI'20) which are missing in the discussion.
+ Some model designs need to be further justified
    - The authors claimed that the proposed online boundary sample detection method is better than directly using the consistency of pseudo labels among neighbours to identify boundary labels, I'd suggest adding a simple ablation study to support this claim, e.g. replace boundary ratio with the consistency of sample's pseudo labels to its neighbours and keep all the other designs unchanged.
    - To add a visualisation of features and highlight those boundary samples identified by the online boundary detection method (e.g. samples with a larger boundary ratio are shown as smaller dots) might be more intuitive for understanding how the data selection is conducted and why it is beneficial
    - as the neighbourhoods are discovered according to the refined features obtained by multiple times of message propagation (Eq.(6)), it will be interesting to see whether the purity of neighbourhoods is positive correlated to the message propagation times, i.e. neighbourhood mined by $h^{(i+1)}$ is more reliable than that by $h^{(i)}$.
    - what's the time complexity of mining contextual affinitive neighbourhood.
    - why not test on Tiny-ImageNet which is another popular benchmark dataset for image clustering and is larger than those datasets adopted here

**Questions:**

Given that the proposed model is trained to produce class-sensitive features of images, it will be interesting to further explore whether such discriminative features can also benefit other downstream tasks beyond image clustering

**Limitations:**

Yes, the authors have addressed the limitation in the conclusion section of the paper.

---

> ### Author Rebuttal · Authors · 2023-08-10
>
> We would like to thank reviewer vt5R for constructive comments on improving our manuscripts, and our responses are listed point-by-point below.
>
> **q1: About clarity of the writing.**
>
> 1) (**Clarifications on model outputs**) Many thanks for giving this actionable suggestion to polish our manuscripts for ease of understanding, we agree that our goal is to learn a clustered representation that better fits k-means for clustering, which is a common paradigm in deep clustering.
> 2) (**Definition of  $h_{i}^{(0)}$**) You are absolutely correct that $h_{i}^{(0)}$ is defined as the feature of vertice $v_i$ constructed in the $k$-NN graph, which will be made clear in our revised version.
> 3) (**The use of $r$**) The $r$ in Eq. 7 is used as a superscript to literally denote the **r**efined features, while $r_i$ in Eq. 11 simply denotes the value of the ratio. The two terms have non-overlapped meanings, and we will change the notation for better clarity.
> 4) (**Message propagation times**) Since there are two identical layers in message propagation, we believe the number of iterations is 2, which will also be made clear in our revised version.
>
> **q2: Related works.**
> Thanks for pointing out these papers that are worth discussing on a concrete basis. Though these papers share similar spirits using neighborhoods as tools, we would like to highlight the critical differences regarding motivation and technical contributions.
> - Our motivation goes beyond exploiting local neighborhoods since many prior works in deep clustering also have implemented this vision. Instead, our major motivation is inspired by the versatile re-ranking methods that excel in image retrieval via post-processing on evaluation sets, and apply their advantages in exploring ConAff neighborhoods in deep clustering.
> - The two mentioned works both build $k$-NN graph based on **local neighborhoods** in an **offline** manner (i.e. over the entire dataset). Specifically, for the latter work PAD, it expands the neighbor relations by searching the strongly-connected subgraphs over the whole dataset, which is orthogonal to our method. Our method, instead, explores ConAff neighborhoods via **online re-ranking** within a batch, sufficiently considering **reciprocal relations**.
>
> **q3: Model Designs.**
> 1) (**Ablation of consistency of pseudo labels**) We appreciate the reviewer's insightful suggestion. Indeed, our experiments have corroborated this observation, as detailed in the ablation study below (in %). From the results, it's evident that online boundary detection (OBD) outperforms the consistency of pseudo labeling (CPL). This aligns with our observation that CPL often struggles to identify boundary samples. This is because their neighborhoods typically possess identical pseudo labels, resulting in a consistent ratio of 1, which unfortunately doesn't capture the true cluster structure.
>   | others+OBD | others+CPL | NMI | ACC | ARI|
>   | :---: | :---: | :---: | :---: | :---: |
>   | | $\checkmark$ | 84.9 | 91.5 | 82.8 |
>   | $\checkmark$ | | 86.7 | 93.2 | 86.1 |
>
>
>
>
>
> 2) **(Visualizations)** We are grateful for the reviewer's insightful points. In response, we have incorporated the suggested visualization in the attached pdf in our general response. As can be observed in Figure 1(a), the overlapped regions of the 2D cluster boundaries predominantly consist of our identified boundary samples. To provide a more focused view, Figure 1(b) exclusively displays these detected boundary samples, effectively highlighting the contours of the clusters.
>
>
> 3) **(Ablation of message propagation times)** In our implementation, the message propagation times are equal to the number of layers, which we empirically set as 2. We further evaluate the purity of ConAff neighborhoods using various numbers of layers on CIFAR-10, as shown below (in %). The result shows that message propagation refines the purity of ConAffN, and the effect is more pronounced with more propagation times. Notably, we kindly note that the quality of a neighborhood is not limited to its purity but also linked to its richness as discussed in L106-108, which affects the final performance.
>   |  num of layers| epoch | purity |
>   | :---: | :---: | :---: |
>   | 0 | 800 | 84.8 |
>   | 1 | 800 | 85.7 |
>   | 2 | 800 | 86.4 |
>
> 4) (**Time Complexity**) Theoretically, the time complexity of ConAffN is $O(n^2) + O(Lk_2n) = O(n^2)$, with n representing the batch size. The constants L and k2 signify the number of layers and expanded neighbors, respectively. However, it's crucial to emphasize that our practical implementation of ConAffN is remarkably efficient. This is evidenced by the training speed comparison with BYOL presented in Table 3 of the Supplementary Material. We attribute this efficiency to the highly parallelizable computation through cuda implementation, which significantly outpaces traditional re-ranking methods.
> 5) (**Why not test on Tiny-ImageNet**) We did evaluate our method on Tiny-ImageNet in the Supplementary Material. We have achieved the best-performing results on Tiny-ImageNet with **46.2 | 30.8  | 18.4**  regarding NMI | ACC | ARI, surpassing the previous methods by at least 3.2% in ARI.
>
> **q4: Downstream applications.**
>
> Thanks for this valuable question, we agree that it is an interesting direction to evaluate ConAff neighborhoods from a broader perspective, hopefully benefiting other downstream tasks. At the same time, we also notice that self-supervised learning focuses more on linear evaluation performance and transferring behavior on detection and segmentation, while our current focus is dedicated to clustering, where extensive experiments (6 benchmarks) with non-trivial performance gains show that our method is self-contained for this line of research. We hope to discover more possibilities of ConAffN in our future work. We would also like to add this discussion to the Appendix.

---

> > ### Comment · Reviewer_vt5R · 2023-08-17
> >
> > I have checked the comments from other reviewers and the authors' responses. Most of my concerns and misunderstanding have been resolved and clarified with supporting materials supplied. I'd like to urge the authors to check the paper thoroughly to make sure all the notations are defined and used coherently without ambiguity. I also agree with other reviewers that there're still several impractical assumptions held in deep clustering, however, I don't think this degraded the technical contributions of this paper and encourage the authors to explore more on those problems which will certainly be another strong contribution for the community. Overall, with all the clarifications and analyses from the authors, the proposed method is sound and impactful to me and I'm happy to change my vote to accept.

---

### Official Review · Reviewer_7yhr · 2023-07-04

**Soundness:** 3 good
**Presentation:** 3 good
**Contribution:** 2 fair
**Rating:** 6
**Confidence:** 5

**Summary:**

This paper improves self-supervised learning for deep clustering. The self-supervision is based on non-contrastive loss functions, which aim to reduce distance between similar samples. Previous algorithms have proposed both the instance level loss functions and group level cost functions. The current paper improves the group level cost function where the group is neighbours of an anchor sample.  The current work proposes a different way to compute neighbours of each anchor sample. It also proposes to clean the neighbours by removing samples at the cluster boundaries. A scheme to proposed to start training with samples closer to the cluster centre as neighbours and then at the later iterations allowing more noisy samples to contribute in the loss computation.

**Strengths:**

1.	This paper is well-written and easy to follow. Most of the details are fully illustrated and the ablation study is quite adequate.
2.	Good empirical results were reported on different clustering benchmarks.
3.	The model structure is carefully designed, and the paper provides a clear explanation of each component along with their respective justifications.
4.	The idea of using k-NN graph for affinitive neighbourhood discovery is simple and efficient.


**Weaknesses:**

1. Motivation of defining a graph as in Eq (3) and then message passing as in Eq (6) is completely missing. Theoretical justification and ablation study to justify both of these steps is essential for this paper.	What if these steps are removed or added with existing group based method?

2. Given the rapid progress in the field, the baseline BYOL used here  is outdated. Compared methods are mostly up to 2021. Many methods have been proposed in 2022 and 2023. Will the proposed steps improve a recent baseline? Also please include some recent methods in comparisons.

3. There is no connection between Eqs (3)-(4) and Eqs (5)-(6). The node representations v_i are never used. Using max in Eq (11) has no advantage because by definition d^N>d^i. In Eq (10) a condition k not equal to \pi(x_i) must be imposed.

4. In experiments, the largest dataset only contains 20 clusters. Will the proposed method be effective on datasets with more categories? Many SOTA methods have shown performance at CIFAR100, therefore authors should give performance for at least one such dataset.

5.	Please analyse the influence of dropping the initial training with BYOL loss on performance.

6.	The backbones are different from other baselines. Please analyse the difference of baselines and its influence on performance.

7.	How to choose values of k1, k2 and fr? Is there any data driven scheme?

8.	In fIgure 3b,  the performance degrades for k2=20. Please provide explanation why is this happening.

9. There is no details of message passing in the paper. Value of alpha and how many message passing iterations are used?

10. Also some terms like `employs an efficient online re-ranking process' and 'encourages cross-view neighborhood consistency' in the conclusion remain unexplained. Where are these terms in the methodology section?  Similarly, in the abstract, following sentence is not clear: 'However, the samples in the local neighborhood may be limited due to their close proximity to each other, which may not provide substantial and diverse supervision signals.'. Similarly following motivation is missing: `Inspired by the
versatile re-ranking methods in the context of image retrieval'. Please improve the abstract to represent the contributions of the paper and remove redundant claims which are not materialised in the paper.


**Questions:**

The same as in Weaknesses section. Authors please respond to the points raised in the weaknesses section.

**Limitations:**

I don't see the proposed algorithm assumes balanced cluster sizes as authors mentioned. The main limitation may be fewer number of classes in the dataset.

---

> ### Author Rebuttal · Authors · 2023-08-10
>
> We really thank reviewer 7yhr for the favorable assessments and constructive suggestions. We would like to address your concerns point-by-point below.
>
> **q1: Motivation of Eq. 3 is missing. Theoretical justification and ablation of ConAff is needed.**
>
> - (**Motivations**) We would like to highlight that defining the graph in Eq (3) and message passing in Eq (6) coincides with our motivation to "employ an efficient online re-ranking process" for the construction of the ConAff neighborhood.
>
> -  (**Theoretical justifications**) At **step 1** (Eq. 3), the $k$-NN graph initializes node features as $k$-reciprocal embeddings, which are essential for re-ranking. Unlike traditional re-ranking, graph construction is more efficient via cuda implementation. At **step 2** (E. 6), the message propagation enables query expansion in re-ranking, spreading node features along with edge weights, also ensuring efficiency.
>
> -  (**Ablation of ConAff**) Per your suggestion, we make a component analysis of ConAffN below (in %). Also, our ablation in Table 2 shows that when ConAffN is completely replaced with the local neighborhood, the result degrades by 3.7% w.r.t. ARI, meaning that ConAffN is essential in this paper.
> | step-1 | step-2 | NMI | ACC | ARI
> | --- | --- | --- | --- | --- |
> | $\checkmark$ | | 83.6 | 90.4 | 81.7 |
> | |$\checkmark$ | 85.1 | 92.1 | 84.6 |
> | $\checkmark$ | $\checkmark$ | 86.7 | 93.2 | 86.1 |
>
> **q2: Will our method improve recent baselines? Include recent methods for comparison.**
>
> - Thanks for this suggestion, we would like to first clarify that the **plug-and-play** role of our approach has been validated on BYOL, SimSiam, and MoCo with non-trivial gains (Table 4 in Supplementary Material) and quick convergence (Fig. 3(c)).
> - We would like to justify all the above SSL frameworks are discussed within the context of instance-aware concordance, either through contrastive or non-contrastive learning, where BYOL is still the de-facto choice. A large portion of recently proposed methods that shine through the SSL field, however, is based on predictive modeling, which is beyond our current paradigm.
> - We further add a recent method DivClust [R1] to enrich our performance comparisons, which are displayed in Table 1 of our attached pdf in general response. We still achieve the best-performing results on most benchmarks.
>
> [R1] DivClust: Controlling Diversity in Deep Clustering, CVPR, 2022.
>
>
> **q3: About formulas of our method.**
> 1) The $v_i$ is denoted as the node feature of the $k$-NN graph in stage-1, and then $v_i$ is used to initialize the input $h_{i}^{(0)}$ of message propagation in stage-2.
> 2) We agree that $max$ in Eq. 11 could be eliminated, though we strictly followed the conventional expression of silhouette score.
> 3) Many thanks for pointing out this typo, we actually aim to find the second nearest cluster in Eq. 10 so that we could detect whether a sample is located near its cluster boundary.
>
> **q4: About performance on datasets with more categories.**
>
> We would like to highlight that the result on Tiny-ImageNet (200 classes, and 100k samples) has been provided in Table 5, Supplementary Material. Concretely, we achieved the best-performing results with 46.2 | 30.8 | 18.4 regarding NMI | ACC | ARI, surpassing the previous methods by at least 3.2% in ARI.
>
> **q5: Analysis of dropping BYOL warm-up.**
>
> We would like to recall that ConNR, as a plug-and-play approach, needs a decent initialization for bootstrapping the cluster structure, therefore, warm-up training with BYOL is necessary. However, this does not devalue our contributions since:
> 1) Warm-up pre-training is not our standalone choice, but a common practice within the existing deep clustering field.
> 2) Our goal is to devise a simple and effective way to better combine the merits of instance-aware and group-aware concordance. Fortunately, we have put forward one feasible solution with non-trivial gains.
>
> **q6: Are the backbones different from other baselines?**
>  We would like to highlight we follow the same setting with previous methods [9, 17, 36, 40] to ensure a fair comparison on each benchmark, which means we have made non-trivial performance gains by the proposed method, instead of some backbone tricks.
>
> **q7: How to choose parameters? Is there a data-driven scheme?**
> Thanks for raising this concern. As stated in L265-267, we set fr=0.7 for all benchmarks; for all the common benchmarks, we set (k1, k2)=(10, 2); for fine-grained benchmark ImageNet-dogs, we set (k1, k2)=(30,10) as fine-grained datasets enjoy more similar semantic information in larger neighborhoods. These values are set empirically rather than in a data-driven scheme.
>
> **q8: Explanations for performance degrading at k2=20.**
> We would like to clarify that k2 denotes the number of expanded neighbors (i.e. the neighbors of reciprocal neighbors), which should be empirically set smaller. This explains why the performance of k2=20 is inferior to k2=10, even though it still outperforms our baseline.
>
> **q9: About details of message passing.**
> We would like to clarify that the $\alpha$ is set to 2, and the number of layers is 2, which means the $k$-reciprocal embeddings in $A$ would be propagated twice.
>
> **q10: About claims in the abstract.**
> Thanks for your suggestion, we would like to refine our claims in the abstract accordingly and address your concerns below.
> 1) The motivation for "online reranking process" could refer to the response in q1.
> 2) The cross-view neighborhood consistency could be explained in the introduction and Figure. 1(a), where t' is pulled towards the neighbors of t. This could be revealed by Eq. 8.
> 3) Maximizing cosine similarity between $x_i, x_j$, which are already very close on the hypersphere, may result in limited supervision signals, shown as small losses during optimization.
> 4) The motivation for re-ranking could refer to  L112-119 and L120-124. Also, versatile re-ranking methods could be viewed in L229-239 in related works.

---

### Official Review · Reviewer_tqX4 · 2023-07-05

**Soundness:** 3 good
**Presentation:** 3 good
**Contribution:** 2 fair
**Rating:** 4
**Confidence:** 4

**Summary:**

The paper tackles the task of deep clustering. The paper introduces a new method, Contextually affinitive Neighborhood Refinery framework (CoNR) that is based on sampling among a relevant set of samples, the contextually affinitive space named ConAff neighborhood. Specifically the method uses top-k re-ranking along with $\alpha$-query extension to select informative neighbours. It further reduces noise among neighbours using a maximum boundary ratio, _i.e._ based on k-means clustering this ratio indicates the proximity of a sample to the cluster boundary. Finally the paper proposes a strategy to progressively relax the condition for sampling, _i.e._ the maximum boundary ratio that is acceptable. The paper validates CoNR on standard deep clustering benchmarks, _i.e._ five relatively small scale dataset: CIFAR-10/20, Imagenet-10/dogs, STL-10. The proposed method shows encouraging results against other deep-clustering state-of-the-art methods. The paper conducts ablation studies of the method on CIFAR-10, and analysis of some hyper-parameters on CIFAR-10 and Imagenet-dogs.

**Strengths:**

S1. The paper is well written and easy to follow.

S2. The CoNR method is well motivated and each component is well introduced.

S3. Experimental results are encouraging when compared to previous state-of-the-art methods for deep clustering.

S4. Ablation studies are present. Ablation studies are really convincing on CIFAR-10: each element seems to participate to the final performance with non-trivial gains.

S5. The paper conducts analysis on the robustness of the method with respect to some of the hyper-parameters used to train CoNR. The method seems robust to variation of hyper-parameters in wide ranges.

S6. Code is present as supplementary material which will help reproducibility. Although I have not tried to use it.

**Weaknesses:**

W1. The paper lacks discussing a closely related paper from the unsupervised metric learning community that uses intra-batch relations and online re-ranking: Self-Taught Metric Learning without Labels, CVPR 2022. The paper should compare results with the STML method.

W2. The paper should discuss relations to other self-supervised method such as ReSSL: Relational Self-Supervised Learning with Weak Augmentation (NeurIPS 2021) and With a Little Help from My Friends: Nearest-Neighbor Contrastive Learning of Visual Representations (ICCV 2021), that use relations with neighbour samples.

W3. The method requires pre-training the network with another self-supervised learning method (BYOL, Bootstrap your own latent: A new approach to self-supervised Learning - NeurIPS 2020). Although this seems to be a common practice in deep-clustering, I believe this reduces the impact of the method, _e.g._ its applicability in large scale setting.

W4. The paper does not conduct experiments on large scale dataset. The two largest datasets in the paper have 60k samples at most. It is not clear whether this type of method will outperform state-of-the-art self-supervised methods methods and k-means clustering on large scale datasets, _e.g._ Imagenet.

W5. The experiments conducted in the paper involved relatively low complexity dataset, _i.e._ with few classes: 20 classes at most. Even though it seems to be, again, a common benchmark, this does not allow to benchmark the method and its effectiveness on complex and realistic tasks.

W6. The paper reports only clustering centered metrics (ACC, NMI and ARI), the paper could also measure common metrics such as knn-accuracy to compare to other method in unsupervised representation learning.

W7. The paper uses the train and test split to train and evaluate the proposed method on the different datasets. The paper should further motivate this beyond it being a common benchmark.

W8: The paper should motivate the use of the 20 subclasses to evaluate CIFAR-100 beyond it being a common benchmark.

**Questions:**

Q1. Line 70: the paper defines the deep clustering task. However it is unclear to me how $k$, the number of clusters, can be set, _i.e._ it is the number of classes on the dataset which should be unknown as it is an unsupervised task. Is there another way to automatically find $k$?

Q2. Eq 10: gives $\pi’$. At the beginning of the epoch will $\pi=\pi’$? The paper could present studies that show the evaluation of $r_i$ during a single epoch and the different epochs.

Q3. Eq 14: gives the final objective for group-aware concordance. Does $\mathcal{N}^{(a)}(I, k)$ include $\mathcal{B}^{(t)}$?

Q4. Line 184: the paper defines $\sigma^{(t)}$ as the boundary ratio, however the boundary ratio was defined in Eq. 11, $r_i$. Maybe $\sigma^{(t)}$ should be referred to as the maximum boundary ratio in line 184.

**Limitations:**

The paper proposes a well motivated and introduced method. The experiments presented in the paper are encouraging.

However the experiments are limited to small scale benchmarks and low complexity datasets. I believe this necessary to compare CoNR to other self-supervised methods (e.g. Dino, Emerging Properties in Self-Supervised Vision Transformers - ICCV 2021) that shines on large scale datasets.
Furthermore the paper should compare the proposed CoNR method to the STML method from Self-taught metric learning without labels (CVPR 2022). Both methods are closely related and I believe this is necessary to validate the CoNR against the state-of-the-art.

---

> ### Author Rebuttal · Authors · 2023-08-10
>
> We sincerely thank reviewer tqX4 for the detailed comments and actionable suggestions. However, especially regarding W4&W5, we would like to kindly note that the concerns about the scalability to large-scale datasets involve a **factual error**.  Our detailed responses are listed as follows.
>
> **q1: Discussion and comparison with STML.**
>
> We would like to add a discussion in our revised version and address our differences in setting, motivation, and technical contributions below.
>
> -  STML mainly targets image retrieval using unsupervised metric learning, which uses different benchmarks, backbones, and baseline methods for comparison.
>
> -  STML is motivated by improving pair-wise similarity via contextual similarity in a teacher-student fashion. By contrast, our motivation is to mine more informative neighbors in ConAffN obtained from propagating a $k$-NN graph.
>
> - STML takes contextual similarity within a pair as a pseudo label, while our method utilizes the ranking statistics of ConAffN within a batch (L39-41, L140-141).
>
> - Per the reviewer's request, we try applying STML to CIFAR-10 using their provided code, as shown below, which does not generalize on our task.
> | Method | NMI | ACC | ARI |
> | --- | --- | --- | --- |
> | STML | 19.9 | 31.3 | 0.17 |
> | Ours | 86.7 | 93.2 | 86.1 |
>
> **q2: Relations to other self-supervised methods, e.g.ReSSL and NNCLR.**
>
> - We would like to highlight that ReSSL is orthogonal to our method, since:
>     - ReSSL obtains a distribution of similarities to the memory bank for each sample, while our method uses intra-batch context to obtain refined feature $h_{i}^{r}$ which is not simply the distribution of similarities but involves reciprocal relations by re-ranking.
>     - ReSSL encourages such distributions of two augmented samples to be similar, while we use the ranking statistics of refined features within a batch to find ConAff neighbors for each sample, and then encourage cross-view neighbor consistency.
>
> - NNCLR, as a pioneering work in self-supervised learning, focuses on mining the nearest neighbor in a local neighborhood, while ConNR takes a step further by mining more informative neighbors in ConAff neighborhoods and combating noises with a progressive strategy.
>
> **q3: The requirement of pre-training downgrades the impact in large-scale settings.**
>
> We respectfully disagree that warm-up training using SSL is a drawback.
> - We did not use pre-training with extra data like ImagNet; more accurately, we utilized warm-up training, which is commonly used in deep clustering.
> - Under the warm-up design, ConNR converges quickly with pronounced gains (Figure 3 (c)), and also serves as a plug-and-play approach. We believe this, in turn, validates the efficacy of our method.
> - There doesn't seem to be a link between the warm-up strategy and scalability to large settings. In fact, our method is tested on Tiny-ImageNet, achieving state-of-the-art performance.
>
> **q4: No large-scale datasets with more than 60k samples?**
>
> We invite the reviewer to check Table 5, Supplementary Material, where we have actually evaluated our method on a large-scale  TinyImageNet, involving **100k samples and 200 classes**. We have achieved the best-performing results on Tiny-ImageNet with **46.2 | 30.8 | 18.4** regarding NMI | ACC | ARI, surpassing the previous methods by at least 3.2% in ARI.
>
> **q5: Low-complexity datasets with no more than 20 classes?**
>
> We actually have reported our method on Tiny-ImageNet containing 200 classes referring to our response in q4. The result table has been provided in our supplementary materials.
>
> **q6:  Lack of knn-accuracy?**
>
> Thanks for your suggestion, however, none of our compared methods has evaluated knn accuracy. We kindly note that i is more often used in self-supervised learning, a different task that focuses more on downstream applications, while our goal in this paper is dedicated to clustered representations.
>
> **q7: Motivate the use of train/test splits on different benchmarks.**
>
> As mentioned in L251, our train/test splits are completely the same with all pre-existing works [9, 17, 16, 20, 23, 24, 36, 38, 39, 40, 44, 54] for a fair comparison. Otherwise, we would not be able to report the results in Table 1.
>
>
> **q8: Motivate the use of CIFAR-20.**
>
> Regarding the use of CIFAR-20, we make a fair comparison with all pre-existing works [9, 17, 16, 20, 23, 24, 36, 38, 39, 40, 44, 54] by using 20 superclasses of CIFAR-100. Otherwise, we would not be able to report the results in Table 1.
>
> **q9: It is unclear how the cluster number K is set.**.
>
> We respectfully note that we adhere to the standard settings used in most deep clustering works, where K is known. Even so, our method remains effective without knowing K in representation learning (Table 2), referring to our responses of q2&q3 to reviewer Prtf.
>
> **q10: Is $\pi'$ equal to $\pi$?**
>
> We would like to point out there is a typo for $\pi'$, which we have fixed in the submitted pdf. In fact, there misses a condition where $k \neq \pi(x_i)$ in Eq. 10, where we actually aim to locate the second nearest cluster for $x_i$.
>
> **q11: Does $N^{a}(i,k)$ include $B^{(t)}$?**
>
> We do not exclude $B^{(t)}$ from $N^{a}(i,k)$ as our aim is to filter out unreliable candidates lying around the boundaries. For each selected candidate, we take its ConAff neighborhood as a whole to preserve its original structure.
>
> **q12: The boundary ratios**
>
> Thanks for raising this concern, $r_i$ is the boundary ratio, and $\sigma^{(t)}$ is a threshold of maximum boundary ratio at epoch $t$.
>
> **q13: About the comparison with DINO shining on large datasets**.
>
> We respectfully note that DINO utilizes a different backbone, ViT. When using the same ViT, DINO's linear evaluation on ImageNet falls short of BYOL, as shown in Table 6, [R1]. Given our method's significant gains over BYOL on Tiny-ImageNet, we could safely believe its efficacy in large-scale settings.
>
> [R1] Self-Supervised Learning via Maximum Entropy Coding, NeurIPS, 2022.

---

> > ### Comment · Reviewer_tqX4 · 2023-08-14
> >
> > I thank the authors for answering some of my questions.
> >
> > (q4&5): However I still have concerns on the scalability, even with the authors pointing to Table 5 in supplementary material with experiments on Tiny-Imagenet, which has _only_ 100k samples and 200 classes as pointed out by the authors. I wonder if the proposed method will work on **large scale** settings, e.g. Imagenet, Google Landmarks v2 etc.
> > As stated in the paper the method is used on top of pre-trained network, so I believe evaluating it on larger datasets will be less computationaly intensive than the pre-training stage (_e.g._ BYOL).
> >
> > (q6-9): On the protocol used for experiments: I believe it would strenghten the experimental validation to present tables with different protocols (using different train / test splits, not have access to K...) Comparison could be done to methods that have published source codes.
> >
> > (q13): Note that Dino was also trained with a ResNet-50. Furthermore I pointed to Dino but would be interested to see more up to date SSL methods for state-of-the-art comparison.

---

> > > ### Author Response · Authors · 2023-08-18
> > > **Author Response for Reviewer tqX4**
> > >
> > > Many thanks for these detailed and valued comments! In your follow-up comments, we notice that: 1) the **experimental settings** you concerned are actually **consistent with the standard setting** in deep clustering; 2) some further questions you raised **are worth exploration but may not degrade our technical contributions as mentioned by reviewer vt5R**. We also agree with reviewer vt5R that **the exploration** of those problems will certainly be **another strong contribution** to the community. Regardless, we'd like to try our best to shed light on these points during the tight discussion period.
> > >
> > > **q4&q5: Concerns on scalability to very large datasets such as ImageNet.**
> > > - We'd like to justify that although our method works on top of a pre-trained model, it still needs further training; thus it is not trivial to conduct experiments on ImageNet.
> > > - We respectfully note that within the deep clustering field, Tiny-Imagenet remains the largest benchmark over most well-known existing works [9, 16, 20, 23, 24, 36, 38, 39, 40, 44, 54],  while ImageNet is still a rare attempt for deep clustering, in contrast to self-supervised learning that focuses on supervised evaluations.
> > > - However, we understand the reviewer's concern that it would be interesting to scale ConNR to extremely large datasets. During the rebuttal periods, we performed a preliminary experiment on ImageNet by using BYOL as our initialization. We trained for 30k steps and evaluated the clustering result on validation sets as shown below. Our method is still adaptable to this very large benchmark, despite the short training schedule given the time constraints of the rebuttal phase.
> > > | Method | NMI | ACC | ARI |
> > > | --- | --- | --- | --- |
> > > | BYOL | 69.3 | 37.6 | 23.2|
> > > | Ours | 70.6 | 38.4 | 23.8|
> > >
> > > **q6-9: Experiments on more evaluation protocols.**
> > >
> > > While **all prior works [9, 16, 17, 20, 23, 24, 36, 38, 39, 40, 44, 54] follow exactly the same protocol for a fair comparison just as we did**, per the reviewer's request, we conduct a series of follow-up experiments for further justifications in terms of different metrics and protocols.
> > > - **(Knn evaluation metric)** We'd like to clarify that knn metric is intrinsically a **supervised setting**, as it utilizes the consensus label among neighboring trained features to infer the label of test features. However, ConNR still gains pronounced improvements on our baseline since it preserves more reliable ConAff neighborhoods, with results shown below.
> > > | Method |  STL10 | ImageNet10 | ImageNetdogs |
> > > | -- |  -- | -- | -- |
> > > | BYOL | 91.8 | 92.8 | 80.0|
> > > | ProPos |  N/A | 93.2 | 85.9 |
> > > | Ours |  93.5 | 94.0 | 87.1 |
> > >
> > > - **(Different train/test splits)**  When test sets are included, it simply means train and test sets are used as a whole for deep clustering without any label information included. We further make experiments for different splits on CIFAR-10 as shown below. The results tell that there is no clear margin between the two settings.
> > > | Method |  NMI | ACC | ARI |
> > > | -- |  -- | -- | --|
> > > | train set only | 87.1 | 93.1 | 86.7 |
> > > | train & test set | 86.7 | 93.2 | 86.1 |
> > >
> > > - **(Not having access to K)** While automatically determining K during clustering is promising, it could be another strong contribution, which may not devalue our current technical contribution since the goal of deep clustering is to find the suitable representation for a pre-defined K. Further, the proposed ConAffN has no reliance on K. Under the setting that does not require K for representation learning, the results are shown below. We report the methods by removing components that require K.
> > > | Method |  NMI | ACC | ARI |
> > > | -- |  -- | -- | -- |
> > > | BYOL |79.4 | 87.8 | 76.6 |
> > > | ProPos |79.4 | 87.9 | 76.4 |
> > > | ConNR | 84.6 | 91.1 | 82.4 |
> > >
> > > **q13: Include DINO for state-of-the-art comparison.**
> > >
> > > Firstly, we'd like to clarify that **the efficacy of self-supervised methods does not necessarily equate to the clustering ability**. While the main objective of self-supervised learning methods is to acquire a generic representation that excels in linear evaluations or a plethora of downstream tasks, deep clustering specifically aims to learn a representation that preserves the cluster structure within the data manifold.
> > >
> > > Secondly, to validate our claim, we have additionally included DINO as another baseline to be combined with our method, following [R1], where the results for CIFAR-10 are shown below.
> > > | Method |  NMI | ACC | ARI |
> > > | --- |  --- | --- | --- |
> > > | DINO | 56.9 | 65.9 | 48.5 |
> > > | DINO+ConNR* | 63.4 | 70.2 | 55.7 |
> > > | BYOL | 79.4 | 87.8 | 76.6 |
> > > | BYOL+ConNR* | 86.7 | 93.2 | 86.1|
> > >
> > > We observe that DINO, which serves as an impactful method in self-supervised learning, unfortunately, has less impact in the deep clustering field compared with BYOL. Even so, our method could still be integrated into DINO with consistent improvements.
> > >
> > > [R1] Solo-learn: A Library of Self-supervised Methods for Visual Representation Learning, JMLR, 2022

---

### Official Review · Reviewer_cuTM · 2023-07-06

**Soundness:** 4 excellent
**Presentation:** 3 good
**Contribution:** 4 excellent
**Rating:** 7
**Confidence:** 5

**Summary:**

In this paper, the authors introduce the Contextually affinitive Neighborhood Refinery (CoNR) framework, which presents a novel approach to deep clustering. CoNR utilizes a simple and efficient online re-ranking process to identify informative neighbors within the ConAff Neighborhood, promoting cross-view neighborhood consistency. Additionally, CoNR incorporates a progressive boundary filtering strategy that effectively tackles intrinsic noises near cluster boundaries. Overall, the CoNR framework offers a promising solution for deep clustering, enabling improved clustering performance and easy integration into existing SSL frameworks.

**Strengths:**

1) The paper introduces a novel framework called Contextually affinitive Neighborhood Refinery (CoNR) for deep clustering, which allows for the mining of more semantically relevant neighbors.

2) One of the key advantages of CoNR is its seamless integration into other SSL frameworks. Its simplicity and effectiveness enable researchers and practitioners to easily incorporate it into their existing frameworks, leading to consistent performance improvements.

3) To tackle the challenge of intrinsic noises near cluster boundaries, the paper presents a progressive boundary filtering strategy. This strategy adopts a self-paced learning paradigm to gradually filter out boundary samples, thereby enhancing the robustness of the clustering process by involving more complex samples.

4) Extensive experiments conducted demonstrate that CoNR surpasses state-of-the-art methods on five widely-used benchmarks, establishing its competitiveness in the field of deep clustering.

**Weaknesses:**

1) The paper does not thoroughly analyze the limitations of the CoNR framework. A deeper exploration of the potential drawbacks or scenarios where CoNR may not perform optimally would have provided a more balanced perspective on its applicability.

2) While the paper briefly mentions the simplicity and effectiveness of CoNR, it could benefit from a more explicit discussion on potential future directions for improvement and extension of the proposed framework

**Questions:**

1) Compared with ConAff neighborhood, why might the local neighborhood samples provide limited supervision signals, and what are the implications of this limitation?
2) How does the proposed method promote cross-view neighborhood consistency? Need more explanation.

**Limitations:**

The potential utilization of the ConAff neighborhood in generic SSL frameworks beyond deep clustering is an interesting prospect. However, the paper does not thoroughly explore this possibility, which can be considered a limitation.

---

> ### Author Rebuttal · Authors · 2023-08-10
>
> We tremendously appreciate reviewer cuTM’s positive assessment and highlighting our contribution to the deep clustering community. In regards to your questions, see our responses below:
>
> **q1: About deeper exploration of limitations of ConNR.**
> Regarding your concerns, we would like to provide a deeper analysis of our limitations and how we could possibly improve them. Though ConNR does not have a strict requirement for balanced classes, it still could not circumvent the **extremely long-tailed** cases, like most existing works. However, we believe that ConAff neighborhoods could serve as a starting point, which could be gracefully combined with a re-sampling strategy of the minority classes to avoid being overwhelmed by the majority classes. On the other hand, we would like to highlight that ConAff could still tolerate a moderately imbalanced case, as samples from minority classes remain contextually similar within the ConAff neighborhood due to their dissimilarity from other samples in the majority classes.
>
> **q2: About future extension and direction of ConNR.**
>
> Thanks for this suggestion, we would like to explicitly add this discussion to our appendix in any revised version.
>
> - Currently, we have validated that ConNR could be easily migrated to various SSL frameworks such as BYOL, SimSiam, and MoCo, all of which are discussed within the context of instance-aware concordance, either through contrastive learning or non-contrastive learning. On the other hand, we notice that there is another line of research in SSL that focuses on **generative (predictive) learning** such as MAE and MixMIM.  We believe it is an interesting direction to further explore ConNR in the generative contexts.
>
> - While our focus in this work is dedicated to clustering, it is possible to also apply our framework beyond our current scope. Specifically, our proposed ConAff neighborhood (ConAffN) is a generic design choice that could be viewed from a broader perspective, hopefully benefiting the task of self-supervised learning that focuses more on downstream applications. Therefore, we believe ConAffN could be further discussed and explored to implement this vision in future works.
>
> **q3: why might local neighborhoods provide limited supervision signals? and the implications?**
>
> Thanks for raising this concern, we would like to elaborate on it more concretely. As discussed in L30-32, the local neighborhoods are considered in a metric space of cosine similarity. Therefore $x_i \in N(x_j)$ indicates that $x_i$ and $x_j$ are already close to each other in terms of cosine similarity, then the supervision signals could be limited when we try to maximize their cosine similarity, manifesting as small losses during optimization. Nevertheless, the proposed ConAff neighborhood considers a contextually affinitive metric space, which makes it able to discover more hard positives, which is better for representation learning.
>
> On the other hand, we invite the reviewer to look at our visualizations in Figure 4 of Supplementary Material, where we have provided a visual display and comparison between the retrieved neighbors produced by local neighborhoods and ConAff neighborhoods. A case in point is shown with sports cars in Figure 4. Given a gray car, the retrieved neighbors of local neighborhoods are mostly in gray. In stark contrast, ConAff neighborhoods involve sports cars with diverse colors, which coincides with our claims.
>
> **q4: How does our method promote cross-view neighborhood consistency?**
>
> Thanks for this question, the cross-view neighborhood consistency could be explained in Figure. 1(a) in an intuitive way: $t'$
>  is pulled towards the ConAff neighbors of $t$ instead of $t'$  itself. Also, Eq. 14 expresses the same idea that the candidate is selected from the online network, while the neighbors of the candidate are selected from the target network. We would like to further explain that such cross-view consistency obeys the principle of instance-aware concordance, and the extraction of the ConAff neighborhood from a target network ensures a stable feature space, as the target network is updated in a momentum scheme.
>
> **q5: The potential possibility of utilization of ConAff neighborhood in generic SSL?**
>
> We would like to agree that it is an interesting direction to explore ConAffN in generic SSL settings, while in this work our major goal is dedicated to learning a clustered and well-separated representation for deep clustering, which is self-contained through extensive experiments. Compared with deep clustering,  we would like to emphasize that self-supervised learning is still a different learning task that focuses on the efficacy of linear evaluation and transferring performance on various downstream tasks. Nevertheless, as shown in our response for **q2**, we agree with the reviewer that it is imperative to explore this possibility, which could be left as our future works.

---

### Official Review · Reviewer_Prtf · 2023-07-06

**Soundness:** 2 fair
**Presentation:** 2 fair
**Contribution:** 3 good
**Rating:** 6
**Confidence:** 3

**Summary:**

This paper proposes a Contextually affinitive Neighborhood Refinery framework for deep clustering which employs an online re-ranking process to mine more informative neighbors and encourage the cross-view neighborhood consistency. A progressive relaxed boundary filtering strategy is present to reduce the influence of neighborhood noise near cluster boundaries.

**Strengths:**

1. The paper is well-written, and the motivation is clear.
2. Multiple baseline methods are included for comparison.


**Weaknesses:**

1. Multiple terms are used without explanation. For example, what does $l$ in Eq. 6 refer to? And $r$ in Eq. 7?
2. Knowing the cluster number is generally an impractical assumption. What if the actual number of classes is unknown? Will the proposed method be able to estimate the number of classes?


**Questions:**

See weakness

**Limitations:**

When class-distribution is imbalanced, the neighborhood could be unreliable for identifying hard positives.

---

> ### Author Rebuttal · Authors · 2023-08-10
>
> We sincerely thank the reviewer for the appreciation of our clear motivation and sufficient experiments. We would like to address your concerns one by one as below.
>
> **q1: Multiple terms are without explanation. What are $l$ and $r$ referring to?**
>
> We would like to try our best to address your concerns regarding the notations you have pointed out. First of all, $l$ simply represents the level of a layer for message propagation, in which the total number of layers is set to 2 (i.e. $L=2$) during the implementation. The message propagation generally takes node features $v_i$ (i.e. $k$-reciprocal encodings) obtained from the previous stage as input $h_{i}^{(0)}$ for the current stage. Since the role of message propagation is to refine those features by local query expansion, we literally use $r$ to represent the final **r**efined features as $h_{i}^{r}$ (as we stated in L121 and L144), which is exactly the output of the final propagation layer $h_{i}^{(L)}$.
>
> As per the reviewer's request, we may use the terms more neatly in our revised version to avoid any confusion.
>
> **q2: Does knowing the cluster number K imply an impractical assumption?**
>
> We would like to first point out that all pre-existing works [9, 17, 16, 20, 23, 24, 36, 38, 39, 40, 44, 54] of deep clustering assume K as a pre-defined number. Therefore, we respectfully disagree that considering K as a known number undermines the value of our work.
>
> Secondly, we would like to clarify that considering the cluster number prior has practical implications.
> - Notably, the current deep clustering field is still dominated by the parametric clustering paradigm since it enjoys flexible scalability and trustworthy performance, despite being confined by a predefined number of clusters that oftentimes could be accessible.
> - Only a few non-parametric deep clustering methods have emerged recently in an attempt to deal with an unknown cluster number. Though this is a promising direction, the current non-parametric deep clustering approaches [R1] mainly implement this vision on small datasets (MINIST and CIFAR-10).
>
> Therefore, we could conclude that both parametric and non-parametric deep clustering methods have their respective applicability and practicality in the mainstream of the deep clustering field.
>
> [R1] Generalised Mutual Information for Discriminative Clustering, NeurIPS, 2022.
>
> **q3:What if the actual number of classes is unknown? Will our method be able to estimate the cluster number?**
>
> While we make a fair comparison with former works by using a known number of clusters, our method could still gracefully handle an unknown number of classes during representation learning, since one of our contributions ConAff neighborhood is not reliant on the cluster number K. This could be evidenced by our ablation study (Table 2). Specifically, when removing the progressive boundary filtering strategy that relies on K, ConNR achieves 84.6 | 91.1 | 82.4 in terms of NMI | ACC | ARI, which still outperforms our baseline by at least +3.3% w.r.t. ACC.
>
> Based on the aforementioned facts, our ConAffN could be simply integrated with existing techniques to automatically select the optimal K among some candidates, such as silhouette analysis, which evaluates whether data points are well matched to their assigned clusters.
>
> **q4: The neighborhood could be unreliable when the class-distribution is imbalanced**.
>
> We would like to emphasize that our method, in comparison to other neighborhood-based methods, can handle imbalance issues to a certain extent. As detailed in our methodology (Lines 105-108), the ConAffN method aims to improve neighborhood quality by focusing on two aspects: cleanliness and richness. Importantly, the imbalance between various classes is unlikely to significantly impact these two aspects. Regarding the ConAff neighborhood in particular, it's worth noting that samples from minority classes remain contextually similar within the ConAff neighborhood due to their dissimilarity from other samples in the majority classes.
>
> We recognize that clustering minor classes could be challenging, particularly in situations of extreme class imbalance. Consequently, we acknowledge this extreme imbalance scenario as a limitation of our approach. We anticipate that there is potential to enhance ConNR through various means, including the exploration of techniques such as re-sampling minor classes in our future research endeavors.
>
> .

---

### Author Rebuttal · Authors · 2023-08-10

# General Response:
We thank the reviewers for their careful reading of our paper, detailed comments, and the help with improving our manuscript. We sincerely appreciate that you have found our work:
- proposes a novel framework that offers a promising solution for deep clustering (Reviewer cuTM)

- proposes really convincing (tqX4) and clearly explained (Reviewer 7yhr, tqX4) components, with adequate justifications (Reviwer 7yhr), comparisons (Prtf), and discussions of the underlying ratinoale (Reviewer vt5R).

- is simpe and effective (Reviewer cuTM, 7yhr, vt5R), enabling seamless integration into existing frameworks (Reviewer cuTM)

- ensures efficiency (Reviewer 7yhr, cuTM) for affinitive neighborhood discovery.

- establishes a new state-of-the-art for image clustering (Reviewer vt5R, 7yhr, cuTM, tqX4)

- is presented with codes to help reproducibility (Reviewer tqX4)

In the subsequent sections, we have addressed all the concerns and questions reviewers raised point-by-point during the rebuttal periods. We have also added a pdf attachment below to involve more figures targeted to specific responses.

We would appreciate it if reviewers could please have a look at our diligence and finalize their assessments, hopefully, more positively. We trust the reviewer and AC discussion would arrive at an informed and fair decision, and we would like to show our gratitude again for the valued feedback and time invested in this process.

Best,

Paper 6511 Authors

---

### Decision · Program_Chairs · 2023-09-21

**Decision:**

Accept (poster)

**Comment:**

This paper got accept recommendation from most of the reviewers. All the reviewers agree that this paper presents a novel method with clear motivation, the experiment results also validate effectiveness of the proposed methods. Some reviewers share concerns on the scalability of the method, results on more complicated datasets. The authors provided detailed responses and addressed these questions. Most of the reviewers are satisfied with this work. AC agrees with them.